# Loss Landscape Degeneracy and Stagewise Development in Transformers

**Jesse Hoogland**[=]                                    *jesse@timaeus.co*
*Timaeus*

**George Wang**[=]                                       *george@timaeus.co*
*Timaeus*

**Matthew Farrugia-Roberts**                             *matthew@far.in.net*
*Department of Computer Science, University of Oxford*

**Liam Carroll**                                         *lemmykc@gmail.com*
*Independent*

**Susan Wei**                                            *susan.wei@monash.edu*
*Department of Econometrics and Business Statistics, Monash University*

**Daniel Murfet**                                        *daniel.murfet@gmail.com*
*School of Mathematics and Statistics, the University of Melbourne*

**Reviewed on OpenReview:** *https://openreview.net/forum?id=45qJyBG8Oj*

## Abstract

Deep learning involves navigating a high-dimensional loss landscape over the neural network parameter space. Over the course of training, complex computational structures form and re-form inside the neural network, leading to shifts in input/output behavior. It is a priority for the science of deep learning to uncover principles governing the development of neural network structure and behavior. Drawing on the framework of singular learning theory, we propose that model development is deeply linked to degeneracy in the local geometry of the loss landscape. We investigate this link by monitoring loss landscape degeneracy throughout training, as quantified by the local learning coefficient, for a transformer language model and an in-context linear regression transformer. We show that training can be divided into distinct periods of change in loss landscape degeneracy, and that these changes in degeneracy coincide with significant changes in the internal computational structure and the input/output behavior of the transformers. This finding provides suggestive evidence that degeneracy and development are linked in transformers, underscoring the potential of a degeneracy-based perspective for understanding modern deep learning.

## 1 Introduction

A striking phenomenon in modern deep learning is the sudden shift in a model's internal computational structure and associated changes in input/output behavior (e.g., Wei et al., 2022; Olsson et al., 2022; McGrath et al., 2022). As large models become more deeply integrated into real-world applications, understanding this phenomenon is a priority for the science of deep learning.

A key feature of the loss landscape of neural networks is *degeneracy*—parameters for which some local perturbations do not affect the loss. Motivated by the perspectives of singular learning theory (SLT; Watanabe,

---

[=]Equal contribution

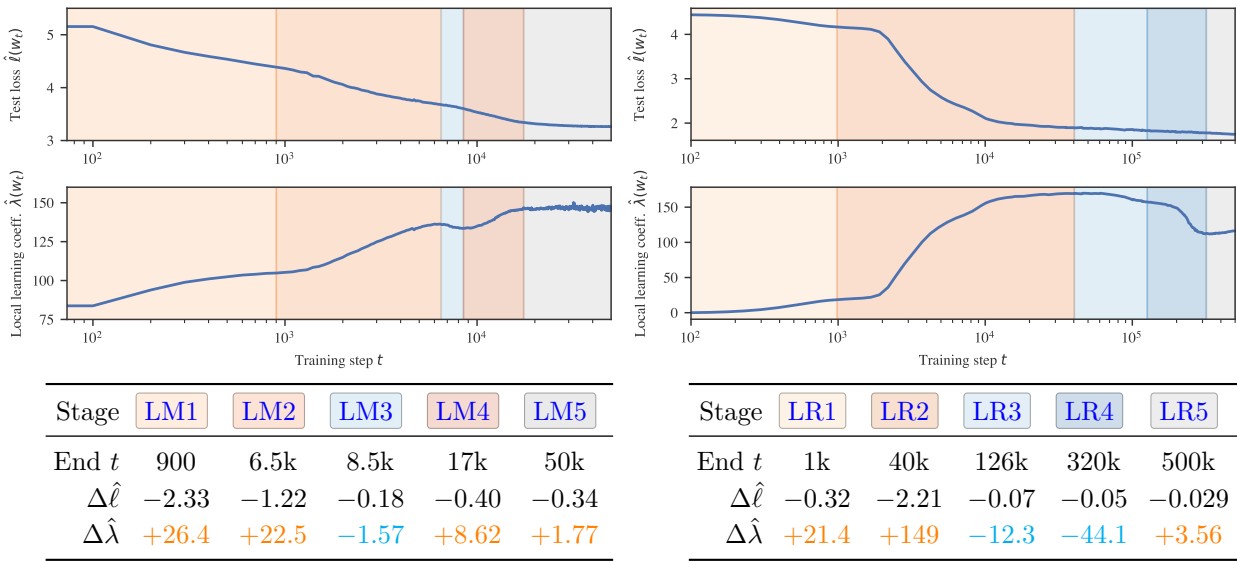

(a) Two-layer attention-only language transformer (LM).

(b) In-context linear regression transformer (LR).

Figure 1: **Tracking loss landscape degeneracy reveals developmental stages.** We train transformer models on both (a) natural language data and (b) synthetic in-context linear regression data. In addition to test loss (top row), we track loss landscape degeneracy as quantified by the local learning coefficient (LLC) (middle row; Section 4). Critical points in the LLC curve mark boundaries between distinct *developmental stages* (bottom row; warm hues for increasing LLC, cold for decreasing LLC; Section 5). We show in Sections 6 and 7 that most of these stages coincide with the formation of significant internal structures or changes in input/output behavior. The language model first learns to predict using bigram statistics (LM1), then common *n*-grams (LM2), before forming the induction circuit studied by Olsson et al. (2022) (LM3&LM4). The regression model first learns the optimal context-independent solution (LR1), then acquires robust in-context learning (LR2), then specializes to the pre-training distribution (LR3&LR4). These stage divisions and interpretations are specific to the above training runs, but we show in Appendix B.4 that similar divisions arise with different training seeds.

2009) and nonlinear dynamics (Waddington, 1957; Thom, 1972), where degeneracy plays a fundamental role in governing development, we believe that studying degeneracy in the local geometry of the loss landscape is key to understanding the development of structure and behavior in modern deep learning.

In this paper, we contribute an empirical investigation of the link between degeneracy and development for transformers in two learning settings. We track loss landscape degeneracy along with model structure and behavior throughout training, using the following methodology.

1. **Transformer training** (Section 3)**:** We train two transformers, a *language model* (LM) with around 3M parameters trained on a subset of the Pile (Gao et al., 2020; Xie et al., 2023), and an *in-context linear regression model* (LR) with around 50k parameters trained on synthetic regression data following Garg et al. (2022).

2. **Degeneracy tracking** (Section 4)**:** We quantify loss landscape degeneracy throughout training by estimating the *local learning coefficient* (LLC; Lau et al., 2025), a measure of degeneracy derived from SLT.

3. **Degeneracy-based stage division** (Section 5)**:** Motivated by the singular learning process in Bayesian inference (Watanabe, 2009, §7.6; Chen et al., 2023), we use critical points in the LLC curve to divide training into approximate *developmental stages*.

4. **Developmental analysis** (Sections 6, 7)**:** We track shifts in each model's internal computational structure and input/output behavior across training, quantified using various setting-specific metrics.

Crucially, we discover that most of the developmental stages identified by changes in loss landscape degeneracy coincide with significant, interpretable shifts in the internal computational structure and input/output behavior of the transformers, showing that the stage division is meaningful. Our investigations are motivated by the hypothesis of a fundamental link between degeneracy and development in deep learning. This hypothesis is theoretically grounded in SLT but so far not empirically validated except in toy models (Chen et al., 2023). We view the above discoveries as preliminary evidence for this hypothesis in larger models, and an indication of the potential of degeneracy as a lens for understanding modern neural network development. Section 8 discusses this and other implications of our investigation.

## 2 Related work

**Degeneracy and development in singular learning theory** Our hypothesis that degeneracy and development are fundamentally linked is motivated by singular learning theory (SLT; Watanabe, 2009), a framework for studying *singular* statistical models (a class that includes neural networks, Hagiwara et al., 1993; Watanabe, 2007; Wei et al., 2023). SLT proves that in singular models, Bayesian inference follows the *singular learning process*, in which degeneracy in the likelihood governs stagewise development in the posterior as the number of samples increases (Watanabe, 2009, §7.6; Lau et al., 2025; Chen et al., 2023). While there are many differences between Bayesian inference and modern neural network training, an analogy to the singular learning process informs our methodology for stage division.

**Degeneracy and development in nonlinear dynamics** Further motivation for our hypothesis comes from viewing neural network training as a stochastic dynamical system, in which the population loss is a governing potential encoding the data distribution. It is well-understood in nonlinear dynamics that degeneracy in the local geometry of a potential can give rise to stagewise development of system structure (Waddington, 1957; Thom, 1972, cf. Franceschelli, 2010). This connection has been observed in biological systems at significant scale and in the presence of stochasticity (Freedman et al., 2023). We emphasize changes in degeneracy over a stage whereas in bifurcation theory the focus is more on the degeneracy at stage boundaries (Rand et al., 2021; MacArthur, 2022; Sáez et al., 2022).

**Stagewise development in deep learning** The idea that neural networks development occurs in stages goes back decades (Raijmakers et al., 1996) and has received renewed attention in modern deep learning (e.g., Wei et al., 2022; Olsson et al., 2022; McGrath et al., 2022; Odonnat et al., 2024; Chen et al., 2024; Edelman et al., 2024). In the case of deep linear networks, we understand theoretically that models learn progressively higher-rank approximations of their data distribution (see, e.g., Baldi & Hornik, 1989; Rogers & McClelland, 2004; Saxe et al., 2019) throughout training. Our findings suggest that studying degeneracy could help generalize this understanding to modern architectures that exhibit more complex internal computational structure, such as transformers.

**Studying loss landscape geometry** Given the central role played by the loss landscape in deep learning, it is unsurprising that there have been many attempts to study its geometry.

One approach is to visualize low-dimensional slices of the loss landscape (Erhan et al., 2010; Goodfellow et al., 2014; Lipton, 2016; Li et al., 2018; Tikeng Notsawo et al., 2024). Unfortunately, a random slice is with high probability a quadratic form associated to nonzero eigenvalues of the Hessian and is thus biased against geometric features that we know are important, such as degeneracy (Wei et al., 2023). Moreover, Antognini & Sohl-Dickstein (2018) have emphasized the difficulty of probing the loss landscape of neural networks with dimensionality reduction tools.

Other standard methods of quantifying the geometry of the loss landscape, such as via the Hessian, are insensitive to important aspects of degeneracy. For example, the Hessian trace or maximum eigenvalues quantify the curvature of a critical point but ignore degenerate dimensions, and the Hessian rank counts the number of degenerate dimensions but fails to distinguish between dimensions by the *order* of their degeneracy (e.g., quartic vs. zero). In contrast, the LLC is a principled quantitative measure of loss landscape degeneracy. Appendix B.5 includes experiments showing that Hessian statistics do not reveal the clear stage boundaries revealed by the LLC in our in-context linear regression setting.

## 3 Training transformers in two settings

We study transformers trained in two learning settings, namely *language modeling* and *in-context linear regression*. These settings have been the subject of recent work on the emergence of in-context learning (ICL), a compelling example of a sudden shift in a model's internal computational structure in modern deep learning (Olsson et al., 2022).

In this section, we describe both settings and introduce their loss functions and data distributions. Common to both settings is a transformer model denoted $f_w$ with parameters $w$, which takes as input a sequence of tokens, also called a *context*. We describe specific architecture details and training hyperparameters in Appendices F.1 and F.2.

**Language modeling**  Elhage et al. (2021) and Olsson et al. (2022) observed that two-layer attention-only transformers (transformers without MLP layers) form interesting internal computational structures supporting ICL, including induction heads. In order to compare with their behavioral and structural analysis we adopt the same architecture. In Appendix E we also study one-layer attention-only transformers. We note that, while we don't study language models with MLP layers (following prior work), we do use MLP layers for in-context linear regression.

We consider the standard task of next-token prediction for token sequences taken from a subset of the Pile (Gao et al., 2020; Xie et al., 2023). We denote the input context by $S_K = (t_1, \ldots, t_K)$ where $K$ is the context length. We denote by $S_{\leq k}$ the prefix context $(t_1, \ldots, t_k)$ of context $S_K$. Our data is a collection of length-$K$ contexts, $\{S_K^i\}_{i=1}^n$. Thus $S_{\leq k}^i$ denotes a prefix of the $i^{\text{th}}$ context, $S_K^i$.

Given the context $S_{\leq k}^i$, the transformer model $f_w$ outputs a vector of logits $f_w(S_{\leq k}^i)$ such that $\text{softmax}(f_w(S_{\leq k}^i))$ is a probability distribution over all tokens (we denote by $\text{softmax}(f_w(S_{\leq k}^i))[t]$ the probability of token $t$). The *per-token empirical loss* for language modeling is then the average cross-entropy between this distribution and the true next token at each index $k \in \{1, \ldots, K-1\}$,

$$\ell_{n,k}(w) = \frac{1}{n}\sum_{i=1}^n -\log\left(\text{softmax}(f_w(S_{\leq k}^i))[t_{k+1}^i]\right). \tag{1}$$

The *empirical loss* is then $\ell_n(w) = \frac{1}{K-1}\sum_{k=1}^{K-1}\ell_{n,k}(w)$, with the *test loss* $\hat{\ell}(w)$ defined analogously on a held-out set of examples. The corresponding *population loss* $\ell(w)$ is defined by taking the expectation with respect to the true distribution of contexts (see also Appendix A.6).

**In-context linear regression**  Following Garg et al. (2022), a number of recent works have explored ICL in the setting of learning simple function classes, such as linear functions. This setting is of interest because we understand theoretically optimal (in-context) linear regression, and because simple transformers are capable of good ICL performance in practice (see, e.g., Garg et al., 2022; Raventós et al., 2023).

We consider a standard synthetic in-context linear regression problem. A *task* is a vector $\mathbf{t} \in \mathbb{R}^D$, and an *example* is a pair $(x, y) \in \mathbb{R}^D \times \mathbb{R}$. We sample a context by sampling one task $\mathbf{t} \sim \mathcal{N}(0, I_D)$ and then sampling $K$ i.i.d. inputs $x_1, \ldots, x_K \sim \mathcal{N}(0, I_D)$ and outputs $y_1, \ldots, y_K \sim \mathcal{N}(\mathbf{t}^\top x, \sigma^2)$. This results in the context $S_K = (x_1, y_1, \ldots, x_{K-1}, y_{K-1}, x_K)$ with label $y_K$. We denote by $S_{\leq k}$ the prefix context $(x_1, y_1, \ldots, x_k)$ of context $S_K$, its label is $y_k$. Appendix F.2.2 describes how we encode the $x_i$ and $y_i$ as tokens. Our data is a set of contexts $\{(\mathbf{t}_i, S_K^i, y_K^i)\}_{i=1}^n$ sampled i.i.d. as described above.

Running a context $S_{\leq k}^i$ through the transformer yields a prediction $\hat{y}_k^i = f_w(S_{\leq k}^i)$, leading to the *per-token empirical loss* for in-context linear regression for $k \in \{1, \ldots, K\}$,

$$\ell_{n,k}(w) = \frac{1}{n}\sum_{i=1}^n (\hat{y}_k^i - y_k^i)^2. \tag{2}$$

The associated *empirical loss* is $\ell_n(w) = \frac{1}{K}\sum_{k=1}^K \ell_{n,k}(w)$. The corresponding *test loss* $\hat{\ell}(w)$ and *population loss* $\ell(w)$ are defined analogously as in the language modeling setting.

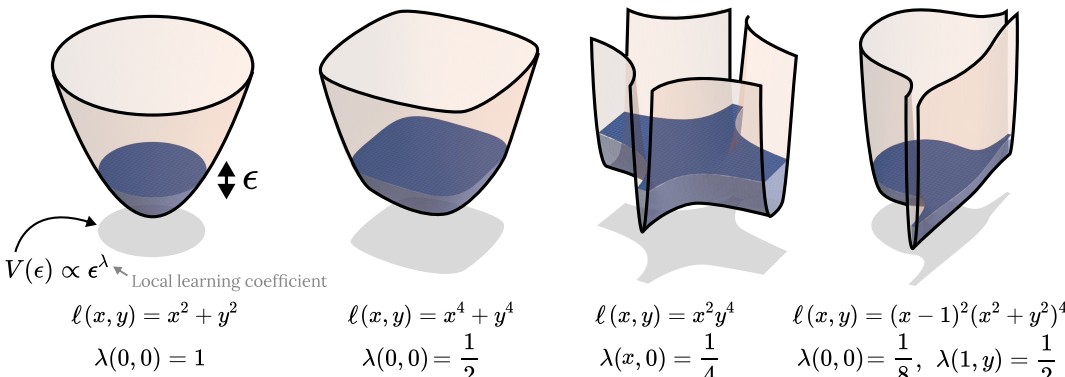

$$V(\epsilon) \propto \epsilon^\lambda$$ Local learning coefficient

$$\ell(x,y) = x^2 + y^2 \qquad \ell(x,y) = x^4 + y^4 \qquad \ell(x,y) = x^2 y^4 \qquad \ell(x,y) = (x-1)^2(x^2+y^2)^4$$

$$\lambda(0,0) = 1 \qquad\qquad \lambda(0,0) = \frac{1}{2} \qquad\qquad \lambda(x,0) = \frac{1}{4} \qquad\qquad \lambda(0,0) = \frac{1}{8}, \ \lambda(1,y) = \frac{1}{2}$$

Figure 2: **The local learning coefficient (LLC) measures loss landscape degeneracy.** The LLC can be defined in terms of the rate at which the parameter space volume (within a given neighborhood and with a given maximum loss) shrinks as the loss threshold is reduced to that of the local minimum. We show four population loss landscapes for a two-dimensional parameter space with decreasing LLC (increasing degeneracy). In these examples, the local multiplicity is 1. See Appendix A.2 for a detailed description of each example, as well as several additional examples.

## 4 Quantifying degeneracy with the local learning coefficient

We track the evolution of degeneracy in the local geometry of the loss landscape throughout training by estimating the local learning coefficient (LLC; Watanabe, 2009; Lau et al., 2025) at model checkpoints. In this section, we review the LLC and the estimation procedure of Lau et al. (2025).

**The local learning coefficient (LLC)** Given a local minimum $w^*$ of a population loss $\ell$ (a negative log likelihood), the LLC of $w^*$, denoted $\lambda(w^*)$, is a positive rational number that measures the amount of *degeneracy* in $\ell$ near $w^*$ (Watanabe, 2009; Lau et al., 2025), i.e., how many ways $w$ can be varied near $w^*$ such that $\ell(w)$ remains equal to $\ell(w^*)$. Formally, the LLC is defined as the *volume-scaling rate* near $w^*$. This is illustrated in Figure 2, further described in Appendix A.1, and treated in full detail in Lau et al. (2025). Informally, the LLC is a measure of minimum "flatness." It improves over conventional (second-order) Hessian-based measures of flatness because the LLC is sensitive to more significant, higher-order contributions to volume-scaling.

**Estimating the LLC** Lau et al. (2025) introduced an estimator for the LLC based on stochastic-gradient Langevin dynamics (SGLD; Welling & Teh, 2011), which we use in our experiments. Let $w^*$ be a local minimum of the population loss $\ell$. The LLC estimate $\hat{\lambda}(w^*)$ is

$$\hat{\lambda}(w^*) = n\beta \left[ \mathbb{E}^{\beta}_{w|w^*,\gamma}[\ell_n(w)] - \ell_n(w^*) \right], \tag{3}$$

where $\mathbb{E}^{\beta}_{w|w^*,\gamma}$ denotes the expectation with respect to the localized Gibbs posterior

$$p(w; w^*, \beta, \gamma) \propto \exp\left\{ -n\beta\ell_n(w) - \frac{\gamma}{2}||w - w^*||_2^2 \right\}$$

with inverse temperature $\beta$ (controlling the contribution of the empirical loss landscape) and localization strength $\gamma$ (controlling proximity to $w^*$). The basic idea behind this estimator is the following: the more degenerate the loss landscape, the easier it is for a sampler exploring the Gibbs posterior to find points of low loss, and, in turn, the lower $\hat{\lambda}(w^*)$. Appendix A.3 discusses technical SGLD details, Appendix A.4 documents the hyperparameters used in our experiments, and Appendix A.5 outlines our hyperparameter tuning procedure.

**Assumptions of LLC estimation** Strictly speaking, the LLC is defined only for loss functions arising as a negative log likelihood, whereas our loss function includes terms from overlapping context prefixes.

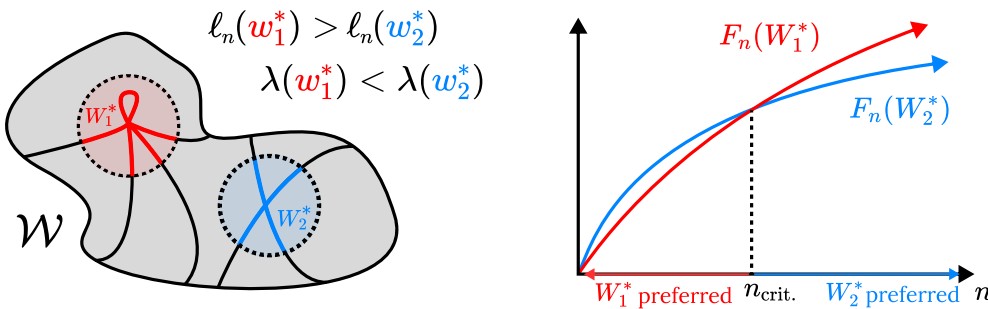

Figure 3: **In the singular learning process, the Bayesian posterior can shift between neighborhoods with different degeneracy.** Watanabe's free energy formula (4) highlights a tradeoff between loss $\ell_n$ (the linear term coefficient) and degeneracy $\lambda$ (the LLC, the logarithmic term coefficient). Consider two local minima $w_1^*, w_2^*$ with neighborhoods $W_1^*, W_2^*$. As the number of samples $n$ increases, if $w_2^*$ has lower loss and higher LLC than $w_1^*$, $W_2^*$ will suddenly achieve lower free energy than $W_1^*$ at some critical sample size $n_{\text{crit}}$, causing the Bayesian posterior to shift from concentrating in $W_1^*$ to $W_2^*$.

It is possible to define a negative log likelihood-based loss for transformer training—we show empirically in Appendix A.6 that this does not have a significant effect on LLC estimates, and so we proceed with overlapping contexts for efficiency.

Moreover, the LLC is defined only for local minima of such loss functions, whereas we note equation (3) is defined for arbitrary $w^*$ and we apply the estimator throughout training. This approach has precedent in prior work on LLC estimation: Lau et al. (2025) showed that when applied to trained parameters, the estimator accurately recovers the learning coefficient associated with a nearby minimum, and Chen et al. (2023) found that the estimator produces reliable results for parameters throughout training. In our case, we obtain stable estimates throughout training given sufficiently strong localization $\gamma$. See Appendix A.7 for more details.

## 5 Degeneracy-based stage division

We use critical points (that is, plateaus, where the first derivative vanishes) in the LLC curve to define *stage boundaries* that divide training into *developmental stages*. This approach is motivated by the singular learning process in Bayesian inference, which we review below.

**Bayesian local free energy**   Let $W^*$ be a neighborhood of a local minimum $w^*$ of the population loss $\ell$ (a negative log likelihood). Given $n$ samples we can define the *local free energy* of the neighborhood (Lau et al., 2025),

$$F_n(W^*) = -\log \int_{W^*} \exp(-n\ell_n(w))\varphi(w)\,dw,$$

where $\varphi(w)$ is a prior positive on the neighborhood $W^*$. The lower the local free energy of a neighborhood $W^*$, the higher the Bayesian posterior mass of $W^*$. In fact, by a log-sum-exp approximation, the Bayesian posterior is approximately concentrated on the neighborhood with the lowest local free energy (cf., Chen et al., 2023).

**The singular learning process**   Watanabe's free energy formula gives, under certain technical conditions, an asymptotic expansion in $n$ of the local free energy (Watanabe, 2018, Theorem 11; Lau et al., 2025):

$$F_n(W^*) = n\ell_n(w^*) + \lambda(w^*)\log n + O_p(\log\log n) \tag{4}$$

Here, $\ell_n(w^*)$ is the empirical loss, $\lambda(w^*)$ is the LLC, and the lower-order terms include a constant contribution from the prior mass of $W^*$.

The first two terms in equation (4) create a tradeoff between accuracy ($\ell_n$) and degeneracy ($\lambda$). Moreover, as $n$ increases, the linear term becomes increasingly important relative to the logarithmic term, changing the

nature of the tradeoff. At certain *critical n* the neighborhood with the lowest local free energy may rapidly change to a neighborhood with *decreased* loss and *increased* LLC, as illustrated in Figure 3.

A sequence of such posterior transitions between increasingly degenerate neighborhoods is a prime example of the *singular learning process* (Watanabe, 2009, §7.6). We note that this is not the only possible dynamic— lower-order terms may also play a role in the evolving competition.

**LLC plateaus separate developmental stages**  While the general connection between the singular learning process in Bayesian inference and stagewise development in deep learning remains to be understood, Chen et al. (2023) showed that, in small autoencoders, both Bayesian inference and stochastic gradient descent undergo rapid transitions between encoding schemes, and these transitions are reflected as sudden changes in the estimated LLC.

This perspective suggests that changes in the loss landscape degeneracy, as measured by the LLC, reflect qualitative changes in the model. In larger models, we expect that these qualitative changes may be more gradual, while still being delineated by brief moments in which the posterior is stably concentrated around a given local minimum. This motivates our approach of identifying *plateaus* in the estimated LLC curve—brief pauses before and after a given increase or decrease in degeneracy—as *stage boundaries* which divide training into approximate *developmental stages*. The resolution of these stage boundaries depends on the density of checkpoints used for LLC estimation and the precision of those estimates.

**Results**  In our experiments, we identify plateaus in the estimated LLC curve by first lightly smoothing the LLC curve with a Gaussian process to facilitate stable numerical differentiation with respect to log training time. We identify plateaus as approximate zeros of this derivative, namely local minima of the absolute derivative that fall below a small threshold (see Appendix B.1). Figure 1 and Appendices B.2 and B.3 show the results. Appendix B.4 shows that similar stage divisions arise for independent training runs.

## 6 Results for language modeling

Plateaus in LLC estimates (Figure 1a) reveal five developmental stages for our language model. In order to validate that this stage division is meaningful, we search for concomitant changes in the model's input/output behavior and its internal computational structure. In this section, we report a range of setting-specific metrics that reveal the following significant, interpretable changes coinciding with each stage: in LM1 the model learns to predict according to bigram statistics; in LM2 the model learns to predict frequent *n*-grams and use the positional embedding; in LM3 and LM4 the model respectively forms "previous-token heads" and "induction heads" as part of the same induction circuit studied by Olsson et al. (2022). Note that we did not discover significant changes in LM5, and we do not claim that these are the only interesting developmental changes occurring throughout training. There may be other interesting developmental changes that are not captured by our metrics, or are not significant enough to not show up in the LLC curve.

### 6.1 Stage LM1 (0–900 steps)

**Learning bigram statistics**  Figure 4(a) shows that the *bigram score*—the average cross entropy between model logits and empirical bigram frequencies (see Appendix C.1.1)—is minimized around the LM1–LM2 boundary, with a value only 0.3 nats above the irreducible entropy of the empirical bigram distribution. This suggests that during LM1 the model learns to predict using bigram statistics (the optimal next-token prediction given only the current token).

### 6.2 Stage LM2 (900–6.5k steps)

**Using positional information**  During LM2 the positional embedding becomes structurally important. Figure 4(b) shows that here the test loss for the model with the positional embedding zero-ablated diverges from the test loss of the unablated model (see Appendix C.2.1). Specifically, we mean setting learned positional embeddings to zero during evaluation. Conditional on our architecture this establishes whether the model effectively uses positional information. A similar method could be used in a model without learned

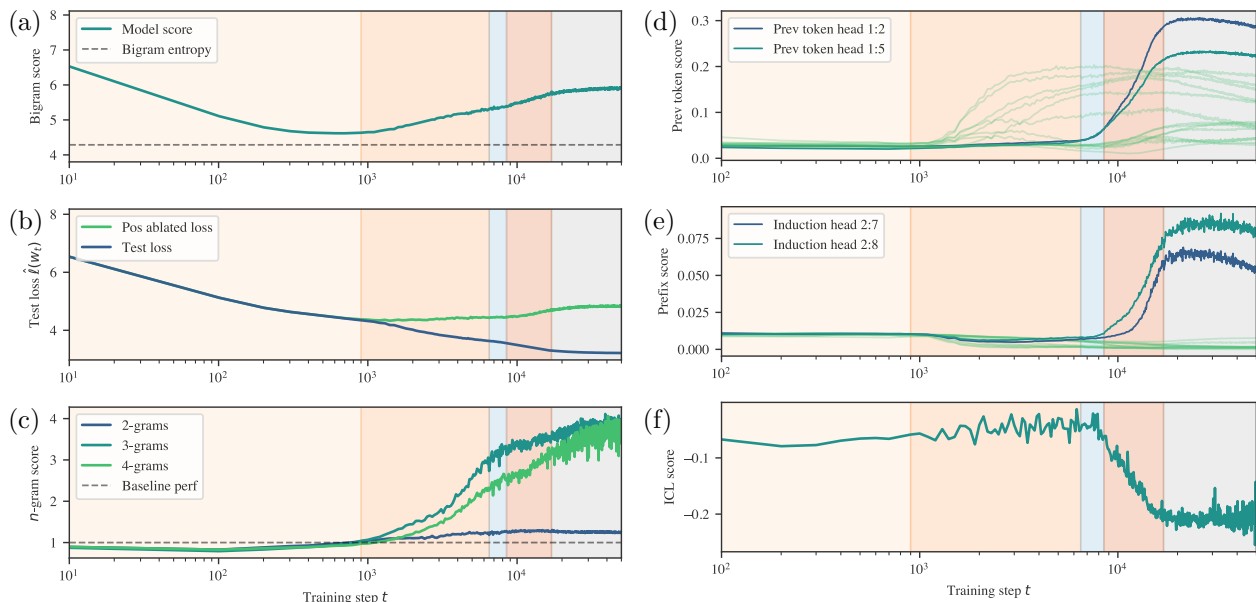

Figure 4: **Language model stages coincide with significant structural and behavioral changes.**
(a) The model learns bigram statistics in LM1, (b) then the positional embedding becomes useful from LM2,
(c) enabling the learning of common *n*-grams. Induction circuit formation begins with (d) previous-token
heads in LM3, followed by (e) induction heads in LM4, leading to (f) a drop in ICL score indicating the
acquisition of in-context learning. Note: in (d,e), *l:h* denotes attention head *h* in layer *l*; dark lines indicate
heads comprising the induction circuit.

positional embeddings. There is also an uptick in previous-token attention among some first-layer attention
heads shown in green in Figure 4(d).

**Learning common *n*-grams**    We define an *n-gram score* as the ratio of final-position token loss on (1) a
baseline set of samples from a validation set truncated to *n* tokens, and (2) a fixed set of common *n*-grams
(see Appendix C.1.2). To compute the "common n-grams" score after extracting the top 1000 n-grams, we
compute the loss on contexts like

$$[\texttt{<bos\_token>, <token\_1>, <token\_2>, ..., <token\_n>}]$$

using the loss on `<token_n>` and normalize against the average loss on the *n*-th token of similar-length
contexts drawn from the pretraining distribution, then divide the *n*-th token loss of truncated pretraining
contexts by the *n*-gram loss to get the *n*-gram score.

Figure 4(c) shows a large improvement in *n*-gram score for $n = 3, 4$ during LM2. This suggests that during
LM2 the model memorizes and learns to predict common *n*-grams for $n > 2$ (note this requires using the
positional encoding and may also involve previous-token heads).

**Foundations of induction circuit**    In this stage, the heads that eventually become previous-token and
induction heads in future stages begin to compose (that is, read from and write to a shared residual stream
subspace; see Figure C.4 and Appendix C.2.2). This suggests that the foundations for the induction circuit
are laid in advance of any measurable change in model outputs or attention patterns.

### 6.3   Stages LM3 & LM4 (6.5k–8.5k & 8.5k–17k steps)

**Formation of induction circuit** (as studied in Olsson et al., 2022)    Figure 4(d) shows the *previous-token
matching score* (Appendix C.2.3) rises over LM3 and LM4 for the two first-layer heads that eventually par-

ticipate in the induction circuit (as distinguished by their composition scores, Appendix C.2.2). Figure 4(e) shows that during LM4 there is an increase in the *prefix-matching score* (Appendix C.2.4) for the two second-layer induction heads that complete the induction circuit. Figure 4(f) shows a corresponding drop in the *ICL score* (Appendix C.1.3) as the model begins to perform in-context learning.

The LLC decreases during LM3, suggesting an increase in degeneracy (a decrease in model complexity). This may be related to interaction between heads. It would be interesting to study this stage further via mechanistic interpretability.

# 7 Results for in-context linear regression

Plateaus in the LLC estimates (Figure 1b) reveal five developmental stages for our in-context linear regression model. We validate that this stage division is meaningful by identifying significant, concomitant changes in the model's structure and behavior: in LR1 the model learns to predict without looking at the context; in LR2 the model acquires a robust in-context learning ability; and in LR3 and LR4 the model becomes more fragile to out-of-distribution inputs. We did not discover significant changes in LR5, nor do we claim this is an exhaustive list of developments.

## 7.1 Stage LR1 (0–1k steps)

**Learning to predict without context** Figure 5(a) shows that the mean square prediction for all tokens $\mathbb{E}[\|\hat{y}_k\|^2]$ decreases during LR1, reaching a minimum of 0.1 (smaller than the target noise $\sigma^2 = 0.125$) slightly after the end of LR1. Similar to how the language model learned bigram statistics in LM1, this suggests the model first learns the optimal context-independent prediction $\hat{y}_k = \bar{\mathbf{t}}^\top x_k$ where $\bar{\mathbf{t}}$ is the mean of the task distribution (zero in this case).

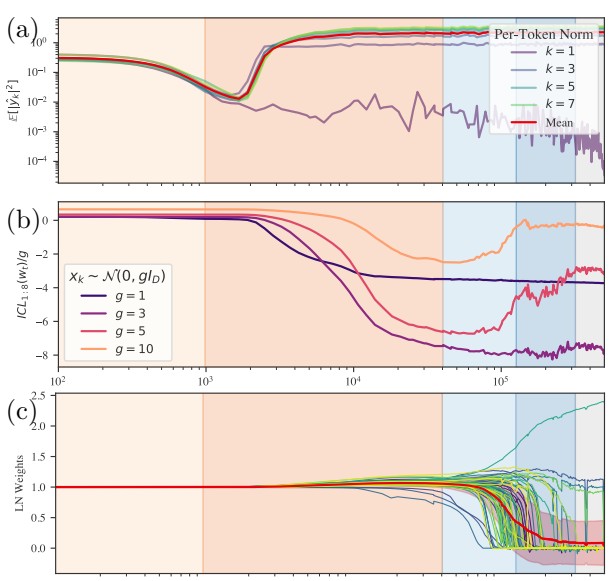

Figure 5: **In-context linear regression model stages coincide with significant structural and behavioral changes.** (a) During LR1, the model learns to make context-independent predictions, $x_k \mapsto \hat{y}_k = 0$. (b) During LR2, ICL performance improves, then during LR3 the model becomes worse at ICL on OOD inputs $x_k \sim \mathcal{N}(0, gI_D)$ for $g > 3$. (c) During LR3 and LR4, layer normalization weights "collapse," possibly contributing to the LLC decrease.

## 7.2 Stage LR2 (1k–40k steps)

**Acquiring in-context learning** Figure 5(b) shows that during LR2 there is a drop in ICL score (Appendix D.1.2), indicating that the model acquires in-context learning.

**Embedding and attention collapse** Appendix D.2 documents additional changes. Near the end of LR2, token and positional embeddings begin to "collapse," effectively losing singular values and aligning with the same activation subspace (Appendices D.2.1 and D.2.2). At the same time, several attention heads form concentrated, input-independent attention patterns (Appendix D.2.3).

## 7.3 Stages LR3 & LR4 (40k–126k & 126k–320k steps)

**Reduced robustness to input magnitude** While performance continues to improve on typical sequences, Figure 5(b) shows that during LR3 and LR4, the model's in-context learning ability *deteriorates* for outlier sequences with higher-than-average $|x_k|$.

**Layer-normalization collapse**   Figure 5(c) shows the individual weights in the final layer normalization module. A large fraction of these weights go to zero in LR3 and LR4. This occurs in tandem with a similar collapse in the weights of the unembedding transforms (Appendix D.2.4). This results in the model learning to read its prediction $\hat{y}_k$ from a handful of privileged dimensions of the residual stream. Since this means that the network outputs become insensitive to changes in many of the parameters, we conjecture that this explains part of the striking decrease in estimated LLC over these stages (Appendix D.2.4).

This collapse is most pronounced and affects the largest proportion of weights in the unembedding module, but in LR4 it spreads to earlier layer normalization modules, particularly the layer normalization module before the first attention block (Appendix D.2.5).

## 8   Discussion

In this paper, we have examined the development of transformer models in two distinct learning settings. We quantified the changes in loss landscape degeneracy throughout transformer training by estimating the local learning coefficient (LLC). Motivated by the singular learning process in Bayesian inference, we divided these training runs into developmental stages at critical points of the LLC curve. We found that these developmental stages roughly coincided with significant changes in the internal computational structure and the input/output behavior of each model. In this section, we discuss several implications of these findings.

**Towards a degeneracy-based understanding of deep learning**   That significant structural and behavioral changes show up in the LLC curve is evidence that the development of our transformers is closely linked to loss landscape degeneracy. This finding underscores the potential of loss landscape degeneracy as a crucial lens through which to study the development of deep learning models.

While we studied two distinct learning settings (including language modeling with a nontrivial transformer architecture), it remains necessary to verify the connection between degeneracy and development across a more diverse range of emergent model structures and behaviors. Moreover, future work should investigate this connection in more depth, seeking to establish a causal connection between changes in degeneracy and changes in structure and behavior.

**Towards developmental interpretability**   We showed that degeneracy can reveal meaningful changes in transformers. We emphasize that our analysis is not exhaustive—we expect only certain "macroscopic" changes, such as the emergence of in-context learning, will have a significant enough effect on loss landscape degeneracy to appear separated by plateaus in the LLC curve. Recent work has extended these ideas by measuring the LLC with respect to network sub-modules and with different data distributions, providing a more refined picture of model development (Wang et al., 2025). We expect this research direction will lead to insights into the development of more complex models.

Loss landscape degeneracy offers a setting-agnostic, "unsupervised" alternative to setting-specific *progress measures* such as those derived by Barak et al. (2022) or developed using mechanistic insights from similar models by Nanda et al. (2023). Both approaches can reveal developments invisible in the loss, but loss landscape degeneracy is able to detect changes without requiring a mechanistic understanding in advance. Of course, once a change is detected through its effect on degeneracy, it remains to interpret the change.

**Cases studies in transformer development**   We do *not* claim that the structural and behavioral developments we observed in each setting are universal phenomena. Transformers trained with different architectures, data distributions, algorithms, or hyperparameters may develop differently. Rather, our detailed analysis contributes two "case studies" to the growing empirical literature on the emergence of structure in transformers.

On this note, our observations extend those of Olsson et al. (2022) and Elhage et al. (2021). We show that before the induction circuit forms, our 2-layer language model learns simpler interpretable strategies (based on bigram statistics and common $n$-grams). This shows that *a single training run* follows a progression akin to that found by Olsson et al. (2022) for *fully-developed models of increasing depth* (they showed that

"0-layer" models learn bigram statistics and 1-layer models learn "skip-trigrams"). A similar progression was observed by Edelman et al. (2024) in a Markovian sequence modeling task.

Moreover, in *both* settings, we saw that before in-context learning emerges, the model learns to predict tokens using the optimal prediction given only the current token (bigram statistics for language modeling, zero for in-context linear regression with this distribution of tasks).

**Development and model complexity**   While we have described the LLC as a measure of loss landscape degeneracy, it can also be understood as a measure of model complexity (cf. Appendix A.2). It is natural for changes in a model's internal structure to show up as a change in complexity. For example, Chen et al. (2024) showed that the emergence of syntactic attention structure coincides with a spike in two model complexity measures, namely the model's *Fisher information* and the *intrinsic dimension* (Facco et al., 2017) of the model's embeddings.

Notably, we observe stages in which the LLC *decreases,* corresponding to a *simplification* of the computational structure of the model. Such model simplification has empirical precedent, for instance with Chen et al. (2024) and the recent literature on grokking (Power et al., 2022; Nanda et al., 2023; Tikeng Notsawo et al., 2024). In our case, the mechanistic nature of the simplification is not fully clear, with the collapse of various weights and attention patterns arising as candidates in the in-context linear regression setting.

This phenomenon is currently not accounted for by theories of neural network development. In the theory of saddle-to-saddle dynamics, deep linear networks learn progressively *more* complex approximations of the data (Saxe et al., 2019). Likewise, the example transitions in the singular learning process outlined in Section 5 and Figure 3 describe LLC *increases.* Though we note that decreasing the LLC while holding the loss constant would be another way to decrease the free energy according to equation (4), providing a full theoretical account of these stages is an open problem.

### Author contributions

All authors communicated regularly about all aspects of the methodology and writing. The following is a non-exhaustive list of some particular areas of individual contribution.

- JH led the coding, experimentation, and analysis for the in-context linear regression setting; assisted with analysis of the language modeling setting; contributed substantially to the writing of the manuscript; and designed the conceptual illustrations.

- GW led the coding, experimentation, and analysis for the language modeling setting; and contributed substantially to the writing of the manuscript.

- MFR led the writing of the current main text, based on earlier versions written with others; and assisted with coding and tuning hyperparameters for the in-context linear regression setting.

- LC assisted with coding, experimentation, and analysis for the in-context linear regression setting.

- SW advised on technical details including assumptions of LLC estimation; and assisted with writing.

- DM had the original idea for the degeneracy-based stage division methodology; contributed substantially to earlier versions of the main text; and coordinated the project.

### Acknowledgments

We thank Edmund Lau for advice on LLC estimation and Mansheej Paul for advice on training transformers in the in-context linear regression setting. We are grateful to Evan Hubinger and Simon Pepin Lehalleur for helpful conversations. For their valuable feedback on the manuscript, we thank Andres Campero, Zach Furman, Simon Pepin Lehalleur, and Nandi Schoots.

We thank Google's TPU Research Cloud program for supporting some of our experiments with Cloud TPUs. JH and GW completed part of this research through the AI Futures Fellowship. MFR completed part of

this research as an independent researcher sponsored by private individuals, part at Timaeus, and part at the University of Oxford. LC's work was supported by Lightspeed Grants.

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

# Appendix

Appendix A reviews the learning coefficient, providing some simple toy examples contrasting the learning coefficient with Hessian-based measures. This section also discusses SGLD-based LLC estimation including experiment hyperparameters (Appendix A.4), and offers a detailed example of the calibrations involved in applying LLC estimation to regression transformers to serve as a reference (Appendix A.5). Appendix B provides further detail on our procedure for LLC-based stage identification, including stages identified in additional training runs and a brief comparison with Hessian statistics. Appendices C and D examine the developmental stages of language models and in-context linear regression in more detail and explain the various metrics we use to track behavioral and structural development. Appendix E describes some additional experiments on a one-layer language model. Appendix F covers transformer training experimental details, such as model architectures, training procedures, and hyperparameters.

To facilitate reproduction of our analyses, we have made our codebase available. A repository containing additional figures and code can be accessed at the URL https://github.com/timaeus-research/icl.

# A  The local learning coefficient (LLC)

## A.1  Formal Definition of the LLC

In the setting of Section 4, let $B$ be a closed ball around $w^*$ such that $w^*$ is a global minimum on $B$, by which we mean a point with (equal) lowest loss. If there are multiple such global minima, the volume asymptotics are determined by the geometry of one that is most degenerate in the precise sense of SLT, formalised in Lau et al. (2025), roughly corresponding to having the lowest LLC. We call this minimum the maximally degenerate global minimum on $B$. Consider the volume of the set of nearby low-loss parameters,

$$V(\epsilon) = \int_B \mathbb{1}\{\ell(w) \leq \ell(w^*) + \epsilon\}\, dw.$$

As $\epsilon \to 0$, $V(\epsilon)$ is asymptotically equivalent to

$$c\epsilon^{\lambda(w^*)} \log(1/\epsilon)^{m(w^*)-1},$$

where $\lambda(w^*)$ is the LLC, $m(w^*)$ is another geometric quantity called the *local multiplicity*, and $c > 0$ is a constant.

## A.2  Interpretations and examples of the LLC

In Section 4, we introduced the LLC as a quantification of geometric degeneracy. In this section, we discuss an additional perspectives on the LLC as a count of the "effective" dimensionality of a parameter, and we give additional examples of the LLC. We refer the reader to Watanabe (2009) and Lau et al. (2025) for more discussion.

The LLC has some similarity to an effective parameter count. If the population loss $\ell$ looks like a quadratic form near $w^*$ then $\lambda(w^*) = \frac{d}{2}$ is half the number of parameters, which we can think of as $d$ contributions of $\frac{1}{2}$ from every independent quadratic direction. If there are only $d-1$ independent quadratic directions, and one coordinate $w_i$ such that small variations in $w_i$ near $w_i^*$ do not change the model relative to the truth (this dimension is "unused") then $\lambda(w^*) = \frac{d-1}{2}$.

The situation becomes more intricate when certain dimensions are degenerate but not completely unused, varying to quartic or higher order near the parameter (rather than being quadratic or flat). While every unused coordinate reduces the LLC by $\frac{1}{2}$, changing the dependency on a coordinate from quadratic ($w_i^2$) to quartic ($w_i^4$) (increasing its *degeneracy* while still "using" it) reduces the contribution to the LLC from $\frac{1}{2}$ to $\frac{1}{4}$.

As a source of intuition, we provide several examples of exact LLCs:

- $\ell(w_1, w_2, w_3) = aw_1^2 + bw_2^2 + cw_3^2$ with $a, b, c > 0$. This function is nondegenerate, and $\lambda(0,0,0) = \frac{1}{2} + \frac{1}{2} + \frac{1}{2} = \frac{3}{2}$. This is independent of $a, b, c$. That is, the LLC $\lambda$ does *not measure curvature*. For this reason, it is better to avoid an intuition that centers on "basin broadness" since this tends to suggest that lowering $a, b, c$ should affect the LLC.

- $\ell(w_1, w_2, w_3) = w_1^2 + w_2^2 + 0$ in $\mathbb{R}^3$ is degenerate, but its level sets are still submanifolds and $\lambda(0,0,0) = \frac{1}{2} + \frac{1}{2}$. In this case the variable $w_3$ is unused, and so does not contribute to the LLC.

- $\ell(w_1, w_2, w_3) = w_1^2 + w_2^4 + w_3^4$ is degenerate and its level sets are, for our purposes, not submanifolds. The singular function germ $(\ell, 0)$ is an object of algebraic geometry, and the appropriate mathematical object is not a *manifold* or a *variety* but a *scheme*. The quartic terms contribute $\frac{1}{4}$ to the LLC, so that $\lambda(0,0,0) = \frac{1}{2} + \frac{1}{4} + \frac{1}{4} = 1$. The higher the power of a variable, the greater the degeneracy and the lower the LLC.

Figure 2 offers several additional examples, from left to right:

- A quadratic potential $\ell_1(w_1, w_2) = w_1^2 + w_2^2$, for which the LLC is maximal in two dimensions, $\lambda_1(0,0) = d/2 = 1$.

- A quartic potential $\ell_2(w_1, w_2) = w_1^4 + w_2^4$, for which the LLC is $\lambda_2(0,0) = 1/2$.

- An even more degenerate potential $\ell_3(w_1, w_2) = w_1^2 w_2^4$, for which $\lambda_3(0,0) = 1/4$. We note that Hessian-derived metrics cannot distinguish between this degeneracy and the preceding quartic degeneracy.

- A qualitatively distinct potential $\ell_4(w_1, w_2) = (w_1 - 1)^2(w_1^2 + w_2^2)^4$ from Lau et al. (2025) with the same LLC at the origin, $\lambda_4(0,0) = 1/4$.

While nondegenerate functions can be locally written as quadratic forms by the Morse Lemma (and are thus qualitatively similar to the approximation obtained from their Hessians), there is no simple equivalent for degenerate functions, such as the population losses of deep neural networks.

### A.3 Estimating LLCs with SGLD

We follow Lau et al. (2025) in using SGLD to estimate the expectation value of the loss in the estimator of the LLC. For a given choice of weights $w^*$ we sample $C$ independent chains with $T_{\text{SGLD}}$ steps per chain. Each chain $c$ is a sequence of weights $\{w_\tau^{(c)}\}_{\tau=1}^{T_{\text{SGLD}}}$. From these samples, we estimate the expectation $\mathbb{E}_{w|w^*,\gamma}^\beta[\mathcal{O}(w)]$ of an observable $\mathcal{O}$ by

$$\frac{1}{CT_{\text{SGLD}}} \sum_{c=1}^{C} \sum_{\tau=1}^{T_{\text{SGLD}}} \mathcal{O}(w_\tau^{(c)}), \tag{5}$$

with an optional burn-in period. Dropping the chain index $c$, each sample in a chain is generated according to:

$$w_{\tau+1} = w_\tau + \Delta w_\tau, \tag{6}$$
$$w_1 = w^*, \tag{7}$$

where the step $\Delta w_\tau$ comes from an SGLD update

$$\Delta w_\tau = \frac{\epsilon}{2} \left( \beta n \nabla \ell_m^{(\tau)}(w_\tau) + \frac{\gamma}{2}(w_\tau - w^*) \right) + \mathcal{N}(0, \epsilon). \tag{8}$$

In each step $\tau$ we sample a mini-batch of size $m$ and the associated empirical loss, denoted $\ell_m^{(\tau)}$, is used to compute the gradient in the SGLD update. We note that LLC estimator defined in (3) uses the expectation $\mathbb{E}^\beta[\ell_n(w)]$ which in the current notation means we should take $\mathcal{O}(w)$ to be $\ell_n(w)$. For computational efficiency we follow Lau et al. (2025) in recycling the mini-batch losses $\ell_m(w_\tau^{(c)})$ computed during the SGLD process. That is, we take $\mathcal{O} = \ell_m^{(\tau)}$ rather than $\mathcal{O} = \ell_n$.

**Time and Space Complexity.** The computational cost per LLC estimate is proportional to a standard training step, denoted $S$. We expect to require a constant number $CT_{\text{SGLD}}$ of samples (on the order of $10^2$–$10^4$) to yield robust estimates, independent of model size. Using logarithmically spaced checkpoints, the total computational complexity for generating an LLC curve over the entire training process scales as $O(SCT_{\text{SGLD}} \log N)$, where $N$ is the total number of training steps. The space complexity incurs a modest linear overhead compared to standard SGD, requiring storage for one additional copy of the weights to enable localization.

### A.4 LLC estimation experiment details

### A.4.1 LLC estimation details for language models

For language models, we use SGLD to sample 20 independent chains with 200 steps per chain and 1 sample per step. For the one-layer model, we used $\epsilon = 0.003, \gamma = 300$, and for the two-layer model we used

$\epsilon = 0.001, \gamma = 100$. Estimating the LLC across all checkpoints took around 200 GPU hours for the two-layer model on a single A100 and around 125 GPU hours for the one-layer model. For additional runs of the two-layer model, we ran fewer chains, bringing the time down to about 2 TPU hours per training run.

We sampled a separate set of 1 million lines (lines 10m-11m) from the DSIR filtered Pile, denoted as $D_{\text{sgld}}$. The first 100,000 lines from this SGLD set (lines 10m-10.1m) were used as a validation set. The sampling of batches for SGLD mirrored the approach taken during the primary training phase. Each SGLD estimation pass was seeded analogously, so, at different checkpoints, the SGLD chains encounter the same selection of batches and injected Gaussian noise.

Table 1: Hyperparameters for estimating the LLC for language models.

| Hyperparameter | Category | Description/Notes | 1-Layer | 2-Layer |
|:---:|:---:|:---:|:---:|:---:|
| C | Sampler | # of chains | | 20 |
| $T_{\text{SGLD}}$ | Sampler | # of SGLD steps / chain | | 200 |
| $\epsilon$ | SGLD | Step size | 0.003 | 0.001 |
| $\gamma$ | SGLD | Localization strength | 300 | 100 |
| $n\beta$ | SGLD | Inverse temperature | | 21.7 |
| $m$ | SGLD | The size of each SGLD batch | | 100 |
| $\mu$ | Data | Dataset size for gradient minibatches | | 13m |

### A.4.2   LLC estimation details for in-context linear regression

For in-context linear regression models, we generate a fixed dataset of $2^{20}$ samples. Using SGLD, we sample 10 independent chains with 5,000 steps per chain, of which the first 1,000 are discarded as a burn-in, after which we draw observations once per step, at a temperature $n\beta = 66.7$, $\epsilon = 0.0003$, and $\gamma = 13.3$, over batches of size $m = 1024$. LLC estimation takes up to 72 CPU-hours per training run.

Table 2: **LLC estimation hyperparameters**. A summary of the hyperparameters involved in estimating the LLC and the default values we use.

| Hyperparameter | Category | Description/Notes | Default Values |
|:---:|:---:|:---:|:---:|
| C | Sampler | # of chains | 10 |
| $T_{\text{SGLD}}$ | Sampler | # of SGLD steps / chain | 5000 |
| $\epsilon$ | SGLD | Step size | 0.0003 |
| $\gamma$ | SGLD | Localization strength | 13.3 |
| $n\beta$ | SGLD | Inverse temperature | 66.7 |
| $m$ | SGLD | The size of each SGLD batch | 1024 |
| $\mu$ | Data | Dataset size for gradient minibatches | $2^{20}$ |

## A.5   A guide to SGLD-based LLC estimation

This section walks through some of the hyperparameter choices and sweeps involved in calibrating LLC estimates. We provide it as a reference for others seeking to adjust LLC estimation to novel settings.

### A.5.1   Varying the temperature

In Lau et al. (2025), the inverse temperature $\beta$ is set to a fixed "optimal" value $\beta^* = 1/\log n$, where $n$ is the number of training samples. In practice, we find that it can be advantageous to sample at a higher temperature.

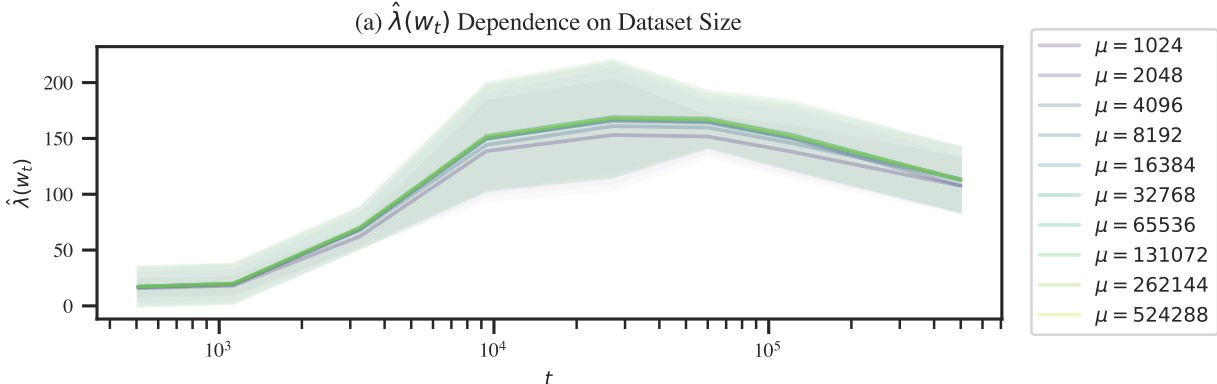

Figure A.1: Past some threshold, the choice of validation set size from which SGLD batches are sampled has little effect on learning coefficient estimates. Estimation hyperparameters are $C = 8, T_{\mathrm{SGLD}} = 2,000, m = 1024, \epsilon = 0.0003, \tilde{\gamma} = 0.01, \tilde{\beta} = 0.01$. Loss is evaluated over gradient minibatches at a representative selection of checkpoints. LLCs quickly converge to a constant value as the size increases.

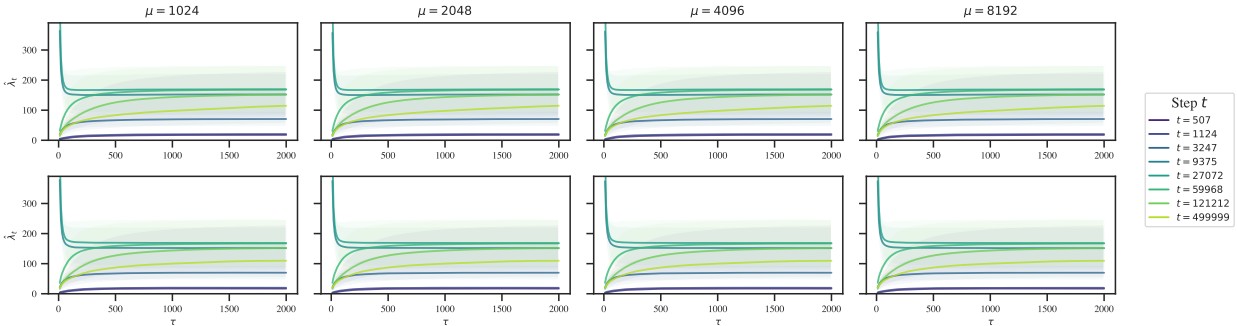

Figure A.2: The size of SGLD minibatches has a negligible effect on LLC estimates (at least among the batch sizes considered). *Top:* Loss is evaluated on the same minibatch as the SGLD gradients. *Bottom:* Loss is evaluated on a newly sampled minibatch from the SGLD gradients (of the same size). Estimation hyperparameters are $C = 8, T_{\mathrm{SGLD}} = 2,000, \mu = 2^{20}$.

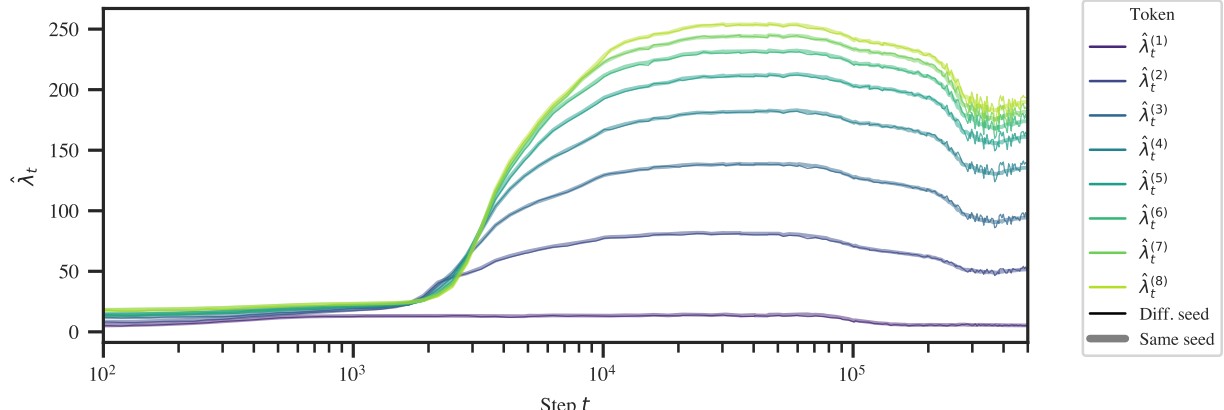

Figure A.3: Consistently seeding SGLD estimates at each checkpoint smooths out the resulting LLC-over-time curve. Except towards the end of training (this is plotted over a log time axis), the difference is barely noticeable. Variable seeds yield a noisier set of estimates.

Since $\beta$ always shows up in a product with $n$ (in (8) for the SGLD step and in (3) for the LLC), we can view the inverse temperature as a multiplier that adjusts the effective size of your dataset. In a Bayesian setting, $\beta = 2$ would mean updating twice on each of the samples in your dataset.

The problem with the default choice of $\beta^*$ is that as we increase $n$ we have to decrease the SGLD step size $\epsilon$ to prevent the update from becoming ill-conditioned, and this eventually causes the gradient term to suppress the noise term. This, in turn, leads to requiring larger batches to suppress the gradient noise and requiring longer chains to sufficiently explore the local posterior (Appendix A.5.3).

Instead of $n\beta = n/\log n$, we perform LLC estimation at $n\beta = m/\log m$, where $m$ is the SGLD batch size.

### A.5.2   Seeding the random noise

To smooth out the $\hat{\lambda}_t$ curves, we reset the random seed before LLC estimation run at each checkpoint. This means the sequence of injected Gaussian noise is the same for LLC estimation runs at different checkpoints. Additionally, if the batch size is held constant, the batch schedule will also be constant across different estimation runs. Figure A.3 shows that this does not affect the overall shape of the learning coefficient curves; it simply smooths it out.

### A.5.3   Calibrating $\epsilon$, $\beta$, and $\gamma$

As a rule of thumb, $\epsilon$ should be large enough that the $\hat{\lambda}$ estimate converges within the $T_{\mathrm{SGLD}}$ steps of each chain but not too large that you run into issues with numerical stability and divergent estimates. Subject to this constraint, $\gamma$ should be as small as possible to encourage exploration without enabling the chains to "escape" to nearby better optima, and $\beta$ should be as large as possible (but no greater than $1/\log n$).

To determine the optimal SGLD hyperparameters, we perform a grid sweep over a reparametrization of the SGLD steps in terms of $\tilde{\beta}, \tilde{\gamma}, \varepsilon$:

$$\Delta w_t = \tilde{\beta}\nabla\ell_m^{(\tau)} + \tilde{\gamma}(w^* - w_t) + \mathcal{N}(0, \varepsilon),$$

where $\tilde{\beta} = \varepsilon\beta n/2$, $\tilde{\gamma} = \varepsilon\gamma/4$.

The results of this hyperparameter sweep are illustrated in Figure A.4 for final checkpoints. Separately (not pictured), we check the resulting hyperparameters for a subset of earlier checkpoints. This is needed since, for example, a well-behaved set of hyperparameters at the end of training may lead to failures like divergent estimates (Figure A.5) earlier in training when the geometry is more complex and thus the chains less stable.

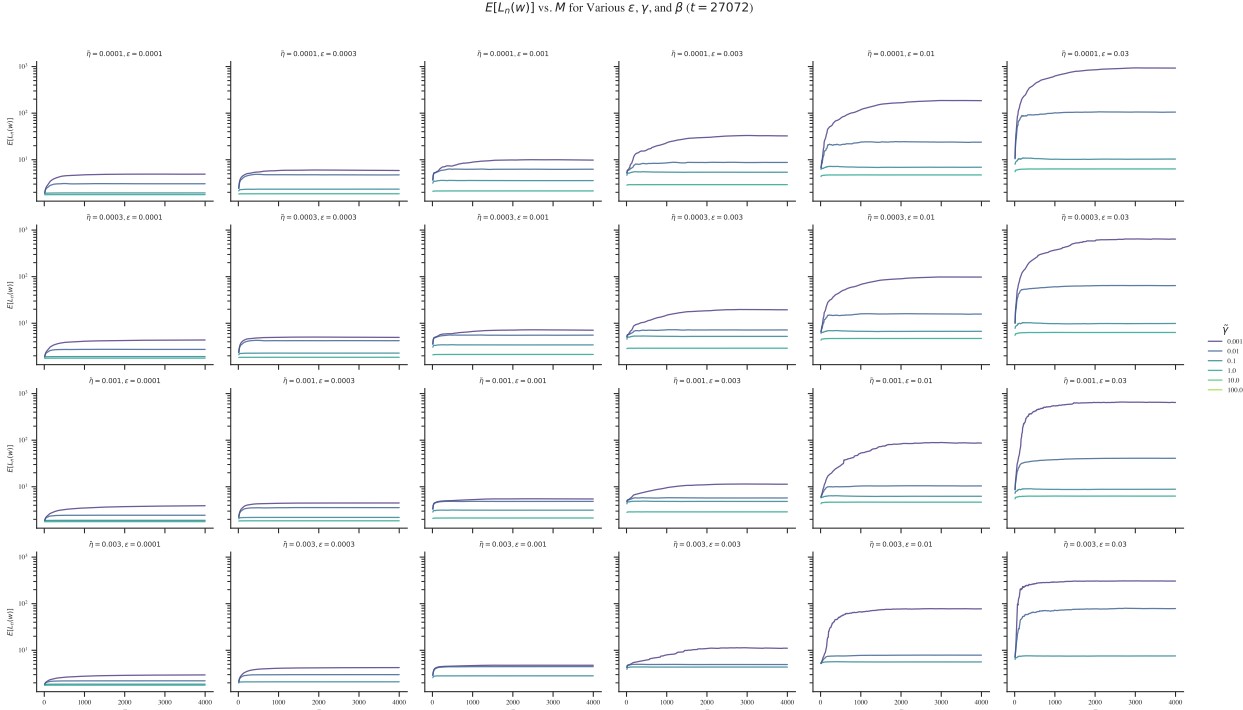

Figure A.4: Results of grid sweep over SGLD hyperparameters for model 0 at $t = 27$k.

### A.5.4 LLC traces

As a useful diagnostic when calibrating the LLC estimates, we propose an online variant for learning coefficient estimation. When overlaid on top of individual-chain LLC traces, this helps reveal common failure modes like divergent estimates, non-converged estimates, and escapes (Figure A.5). These traces display the running estimate of $\hat{\lambda}$ as a function of the number of steps taken in a chain (with the estimate averaged across independent chains).

Define $\hat{\lambda}_\tau(w_0)$, the LLC at $w_0$ after $\tau$ time-steps for a single SGLD chain as follows (Lau et al., 2025):

$$\hat{\lambda}_\tau(w_0) = n\beta \left( \frac{1}{T} \sum_{t=1}^{T} \ell_n(w_\tau) - \ell_n(w_0) \right).$$

Moving terms around, we get,

$$\hat{\lambda}_\tau(w_0) = \frac{n}{\log n} \left( \frac{1}{\tau} \sum_{\tau'=1}^{\tau} \ell_n(w_{\tau'}) - \ell_n(w_0) \right) \tag{9}$$

$$= n\beta \left( \frac{\tau - 1}{\tau} \left( \frac{1}{\tau - 1} \sum_{\tau'=1}^{\tau-1} \ell_n(w_\tau') - \ell_n(w_0) + \ell_n(w_0) \right) + \frac{1}{\tau} \ell_n(w_\tau) - \ell_n(w_0) \right) \tag{10}$$

$$= \frac{\tau - 1}{\tau} \hat{\lambda}_{\tau-1}(w_0) + n\beta \left( \frac{1}{\tau} \ell_n(w_\tau) + \left( \frac{\tau - 1}{\tau} - 1 \right) \ell_n(w_0) \right) \tag{11}$$

$$= \frac{1}{\tau} \left( (\tau - 1)\hat{\lambda}_{\tau-1}(w_0) + n\beta \left( \ell_n(w_\tau) - \ell_n(w_0) \right) \right), \tag{12}$$

where

$$\hat{\lambda}_0(w_0) = 0.$$

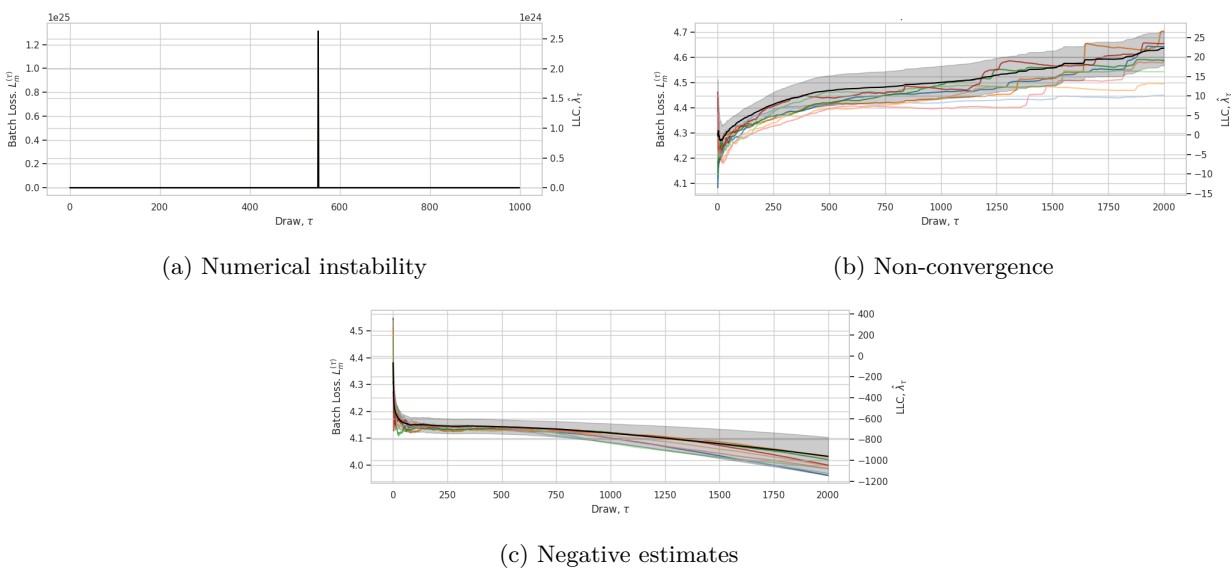

(a) Numerical instability

(b) Non-convergence

(c) Negative estimates

Figure A.5: **Failure modes of SGLD estimation.** *Top left*: the gradient term is too large, leading to issues with numerical instability and exploding $\hat{\lambda}$ estimates. *Top right*: $\epsilon$ is too small, leading to $\hat{\lambda}$ not converging within each chain. *Bottom*: the localization term is too small, which allows the chain to escape to better optima.

This can be easily extended to an online estimate over chains by averaging the update $n\beta\left(\ell_n(w_\tau) - \ell_n(w_0)\right)$ over multiple chains.

## A.6 LLC estimates for a non-log-likelihood-based loss

In the main body, we apply the LLC to empirical loss functions that do not arise as the log likelihood of independent random variables, due to the repeated use of dependent sub-sequences. Here we explain that it is possible to define a proper negative log likelihood over independent observations for the in-context linear regression setting: similar observations can be made in the language modeling setting.

Let $\Pi(k)$ be a probability distribution over the context length $k$. Ideally, the transformer would be trained to make predictions $y_k$ given a context of length $k$ where $k$ is sampled from $\Pi$. With the given distribution over contexts this leads to a negative log likelihood of the form

$$L(w) = \sum_k p_k L_{[k]}(w) \tag{13}$$

where $p_k$ is the probability of sampling $k$ from $\Pi$ and

$$L_{[k]}(w) = \int q(S_k, y_k|\mathbf{t}, k)q(\mathbf{t})\Big[f_w(S_k) - y_k\Big]^2 dS_k \, dy_k \, d\mathbf{t} \tag{14}$$

using the notation of Section 3 so $S_k = (x_1, y_1, \ldots, x_{k-1}, y_{k-1}, x_k)$ is a context of length $k$. It is straightforward to check that this negative log likelihood $L$ agrees with the population loss $\ell$ associated to the empirical loss defined in Section 3. However the empirical quantities $L_n(w)$ and $\ell_n(w)$ defined for a set of samples of size $n$ are *not* the same.

Since we use the empirical loss $\ell_n$ in our calculation of the estimated LLC, whereas the foundational theory of SLT is written in terms of the empirical negative log likelihood $L_n$, it is natural to wonder how much of a difference this makes in practice. Figure A.6 depicts LLC traces (Appendix A.5) for a highlighted number of checkpoints using either a likelihood-based estimate (with variable sequence length) or loss-based estimate (with fixed sequence length). The relative orderings of complexities does not change, and even the values

of the LLC estimates do not make much of a difference, except at the final checkpoint, which has a higher value for the sub-sequence-based estimate.

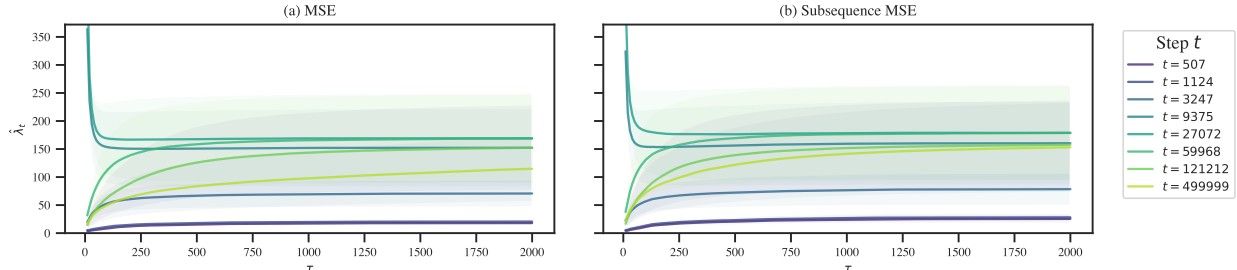

Figure A.6: Loss-based (*left*) and likelihood-based (*right*) LLC estimation yield identically ordered LLC estimates. With the exception of final checkpoint's LLC estimate (which is larger for the loss-based estimate), the values are close to identical. These plots display *LLC traces*, which show the LLC estimate as a function of *SGLD steps*. This is a useful tool for calibrating LLC estimation (Appendix A.5).

### A.7   LLC estimates away from local minima

Our methodology for detecting stages is to apply LLC estimation to compute $\hat{\lambda}(w^*)$ at neural network parameters $w^* = w_t$ across training. In the typical case these parameters will *not* be local minima of the population loss, violating the theoretical conditions under which the LLC is defined.

It is not surprising that the estimator appears to work if $w^*$ is approximately a local minima. Lau et al. (2025) validated their estimator at both parameters constructed to be local minima of the population loss and also at parameters found through training with stochastic gradient descent (possibly not local minima of the empirical loss, let alone the population loss). They showed that in both cases the estimator recovers the true learning coefficient associated with the global minimum of the population loss.

On the other hand, if $w^*$ is far from any local minima, it is *a priori* quite surprising that the SGLD-based estimation procedure works at all, as in this situation one might expect the chains to explore directions in which the loss decreases. Nevertheless, Chen et al. (2023) found that, empirically, LLC estimation away from local minima appears to give sensible results in practice. In our case, with sufficient localization we see stable estimates throughout training.

Theoretically accounting for this phenomenon is an interesting open problem. Perhaps there is a notion of *stably evolving equilibrium* in the setting of neural network training, echoing some of the ideas of Waddington (1957), such that the LLC estimation procedure is effectively giving us the LLC of a different potential to the population loss—a potential for which the current parameter actually *is* at a critical point. We leave addressing this question to future work.

# B LLC-based stage boundary identification

## B.1 Procedure for stage boundary identification

To identify stage boundaries, we look for plateaus in the LLC: checkpoints at which the slope of $\hat{\lambda}(w_t)$ over $t$ vanishes. To mitigate noise in the LLC estimates, we first fit a Gaussian process with some smoothing to the LLC-over-time curve. Then we numerically calculate the slope of this Gaussian process with respect to $\log t$. The logarithm corrects for the fact that the learning coefficient, like the loss, changes less as training progresses. We identify stage boundaries by looking for checkpoints at which this estimated slope equals zero. The results of this procedure are depicted in Figure B.1 for language and Figure B.2 for in-context linear regression.

At a local minima or maxima of the estimated LLC curve identifying a plateau from this estimated slope is straightforward, since the derivative crosses the x-axis. However at a saddle point, the slope may not exactly reach zero, so we have to specify a "tolerance" for the absolute value of the derivative, below which we treat the boundary as an effective plateau.

In this case, we additionally require that the plateau be at a local minimum of the absolute first derivative. Otherwise, we may identify several adjacent points as all constituting a stage boundary.

To summarize, identifying stage boundaries is sensitive to the following choices: the intervals between checkpoints, the amount of smoothing, whether to differentiate with respect to $t$ or $\log t$, and the choice of tolerance. However, once a given choice of these hyperparameters is fixed, stages can be *automatically* identified, without further human judgment.

## B.2 Stage boundary identification details for language model

Figure B.1 displays the test loss and LLC curves from Figure 1a in addition to the weight norm over time and associated slopes. Stage boundaries coincide with where the slope of the LLC crosses zero, that is, where there is a plateau in the LLC.

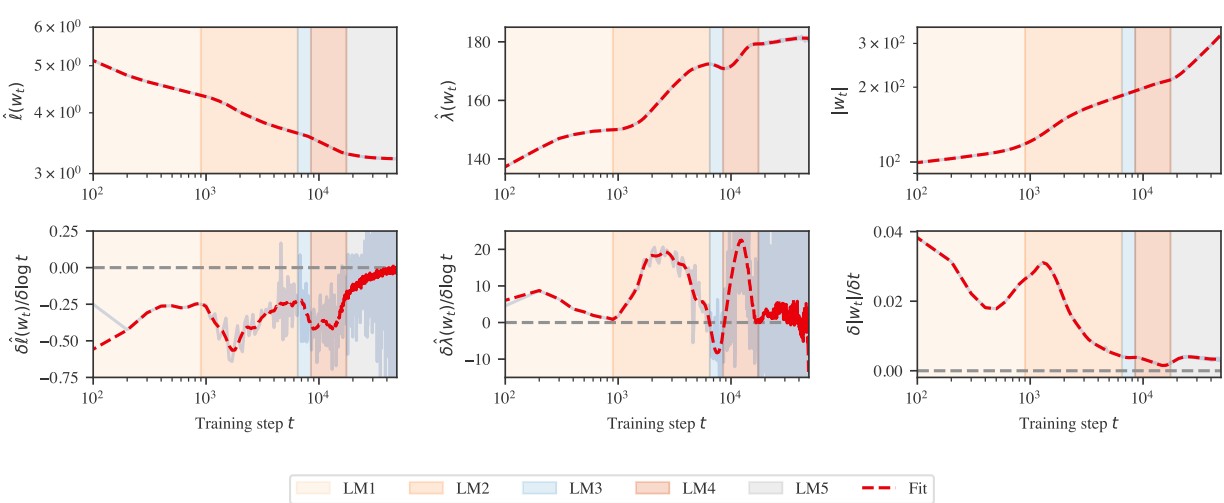

Figure B.1: A more detailed version of Figure 1a for two-layer language models. *Top*: Loss, LLC, and weight norm, along with an overlaid Gaussian process fit to these curves (red dotted lines). *Bottom*: Associated slopes, both numerically estimated finite differences (transparent blue) and of the Gaussian process (red dotted lined). Note that stage LM5 may be subdivided into further stages (Appendix B.1). However, the noise in LLC estimates late in training is high, so we do not draw any conclusions from this.

## B.3 Stage boundary identification details for in-context linear regression

Figure B.2 displays the test loss and LLC curves from Figure 1b in addition to the weight norm over time, and numerically estimated slopes associated to these three metrics. As in the case of language models, we identify stage boundaries by looking for plateaus in the LLC. Unlike the language models, here the boundaries LR1–LR2 and LR2–LR3 are clearly visible in the loss.

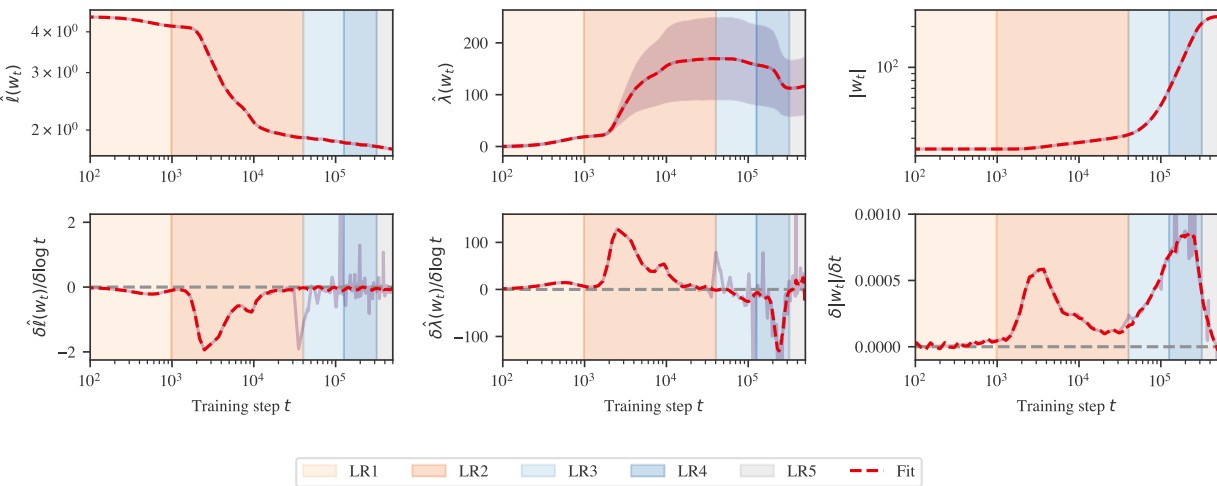

Figure B.2: A more detailed version of Figure 1b for in-context linear regression. *Top*: Loss, LLC, and weight norm, along with an overlaid Gaussian process fit to these curves (red dotted lines). *Bottom*: Associated slopes, both numerically estimated finite differences (transparent blue) and of the Gaussian process (red dotted lined). *Top middle*: Error bars displaying the standard deviation over the 10 SGLD chains are displayed in the background. Note that large error bars across chains are to be expected. Between different SGLD estimations, the variance is much lower. For example, averaged over training, the standard deviation over different seeds is only 4.2.

## B.4 Stage identification for additional training runs

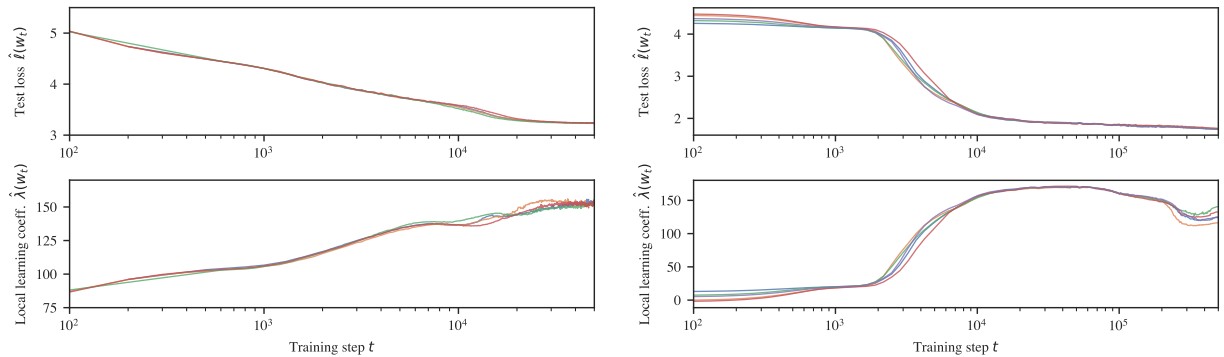

(a) **Two-layer attention-only language transformers.**

(b) **In-context linear regression transformers.**

Figure B.3: Figure 1a and Figure 1b for multiple seeds. In both settings, LLC reveals a consistent set of stages across five seeds. Late-training behavior shows more variance across seeds (see Appendix B.4).

Figure B.3a shows loss and LLC curves for five seeds (differing in model initialization and batch schedule). In each seed, LLC estimation reveals stage LM1–LM4. In three of the five seeds, stage LM5 is subdivided into two additional stages.

Figure B.3b shows loss and LLC curves for five unique seeds (differing in model initialization and batch schedule). In each seed, LLC estimation reveals stages LR1–LR5. There is remarkably little variance across different seeds.

## B.5   Comparison to Hessian statistics

Figure B.4 shows a quantification of the curvature-based notion of flatness captured by the Hessian (in contrast to the degeneracy-based notion of flatness captured by the LLC) for our in-context linear regression transformer. To estimate the trace and maximum eigenvalues shown in this figure, we use the PyHessian library (Yao et al., 2020) over a batch of $m = 1024$ samples.

Crucially, we observe that these Hessian-derived metrics (henceforth, "curvature") and the LLC are *not* consistently correlated. During the first part of LR2, the LLC and the curvature are jointly increasing. Starting at around $t = 20k$, while the LLC is still increasing, the curvature starts decreasing. In the first part of LR3, both metrics decrease in tandem, but as of around $t = 120k$, the curvature turns around and starts increasing.

The Hessian fails to detect three of the four stage boundaries identified by our LLC-based methodology. Since these Hessian-based metrics are dominated by the largest eigenvalues—the directions of maximum curvature—they fail to observe the finer-grained measures of degeneracy that dominate the LLC. Moreover, we observe that LLC estimation is more scalable (empirically, it seems to be roughly linear in parameter count) than estimating the full Hessian (which is quadratic).

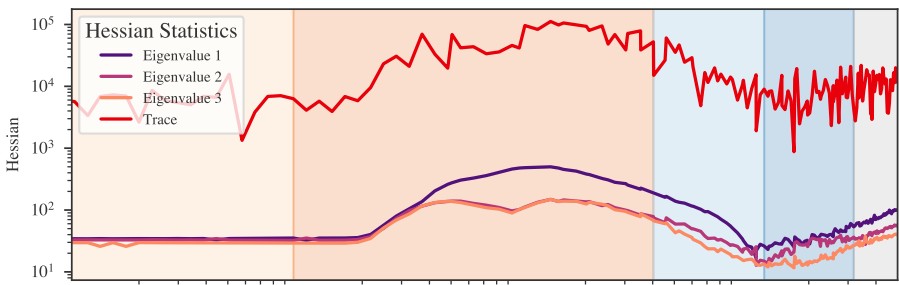

Figure B.4: Hessian-based statistics reveal only one stage boundary in the development of our in-context linear regression transformer.

# C Developmental analysis of language models

In this section, we present further evidence on behavioral (Appendix C.1) and structural (Appendix C.2) development of the language model over the course of training.

## C.1 Behavioral development

### C.1.1 Bigram score

We empirically estimate the conditional bigram distribution by counting instances of bigrams over the training data. From this, we obtain the conditional distribution $\tilde{q}(t'|t)$, the likelihood that a token $t'$ follows $t$. The *bigram score* $B_k^S$ at index $k$ of an input context $S$ is the cross entropy between the model's predictions $p(t_{k+1}|t_k)$ at that position and the empirical bigram distribution,

$$B_k^S = - \sum_{i=1}^{d_{\text{vocab}}} \tilde{q}(t_{k+1}^{(i)}|t_k) \log p(t_{k+1}^{(i)}|t_k), \tag{15}$$

where the $t_{k+1}^{(i)}$ range over the possible second tokens from the tokenizer vocabulary. From this we obtain the *average bigram score*

$$\bar{B} = \frac{1}{n} \sum_{i=1}^{n} B_{k_i}^{S_i}, \tag{16}$$

where we take fixed random sequences of $k_i$ and $S_i$ for $1 \leq i \leq n = 5,000$, which is displayed over training in Figure 4(a). This is compared against the best-achievable bigram score, which is the bigram distribution entropy itself, averaged over the validation set.

### C.1.2 $n$-gram scores

In stage LM2 we consider $n$-grams, which are sequences of $n$ consecutive tokens, meaning 2-grams and bigrams are the same. Specifically, we consider *common $n$-grams*, which is defined heuristically by comparing our 5,000 vocab size tokenizer with the full GPT-2 tokenizer. We use the GPT-2 tokenizer as our heuristic because its vocabulary is constructed iteratively by merging the most frequent pairs of tokens.

We first tokenize the tokens in the full GPT-2 vocabulary to get a list of 50,257 $n$-grams for various $n$. The first 5,000 such $n$-grams are all 1-grams, after which 2-grams begin appearing, then 3-grams, 4-grams, and so on (where 2-grams and 3-grams may still continue to appear later in the vocabulary). We then define the set of common $n$-grams as the first 1,000 $n$-grams that appear in this list for a fixed $n$, $n \geq 2$.

If we track the performance on $n$-grams and see it improve, we may ask whether this is simply a function of the model learning to use more context in general, rather than specifically improving on the set of $n$-grams being tracked. We measure performance against this baseline by defining an *$n$-gram score*. For a fixed $n$, we obtain the average loss $\ell_{\text{gram}}^n$ of the model on predicting the final tokens of our set of 1,000 $n$-grams and also obtain the average loss $\ell_{\text{test}}^n$ of the model on a validation set at position $n$ of each validation sequence. The $n$-gram score is then defined to be $\ell_{\text{test}}^n / \ell_{\text{gram}}^n$.

### C.1.3 In-context learning score

The *in-context learning score* is a behavioral measure of the relative performance of a model later in a sequence versus earlier in the sequence. We define $\text{ICL}_{k_1:k_2}$ to be the loss on token $k_2$ minus the loss on token $k_1$, so a more negative score indicates better relative performance later in the sequence. A more negative ICL score does not, however, mean that a model is achieving better overall loss on later tokens; it is only about the relative improvement. For the language model, we follow a similar construction as Olsson et al. (2022), where we take $k_2$ to be the 500th token and $k_1$ to be the 50th token. This is then averaged over a 100k-row validation dataset. The performance of the language model over the course of training can be seen at the bottom of Figure 4(f).

### C.1.4   Visualizing behavioral changes

In Figure C.1, we visualize changes in the model's input/output behavior by comparing model predictions before and after developmental stages and highlighting tokens with the greatest differences.

(a) LM1 (0 - 900)

<|endoftext|>I should like, before proceeding further, to tell you how I feel about the State which we have described. I might compare myself to a person who, on beholding beautiful animals either created by the painter's art, or, better still, alive but at rest, is seized with a desire of seeing them in motion or engaged in some struggle or conflict to which their forms appear suited;

(b) LM2 (900 - 6,500)

<|endoftext|>In the midst of unexpected circumstances with Linux and Python, the honorable Supreme Court in Boston delivered a ruling emphasizing a crazy database framework last week.

(c) LM3 + LM4 (6,500 - 17,000)

<|endoftext|>Mr. and Mrs. Dursley, of number four, Privet Drive, were proud to say that they were perfectly normal, thank you very much. They were the last people you'd expect to be involved in anything strange or mysterious, because they just didn't hold with such nonsense. Mr. Dursley was the director of a firm called Grunnings, which made drills. He was a big, beefy man with hardly any neck, although he did have a very large mustache. Mrs. Dursley was thin and blonde and had nearly twice the usual amount of neck, which came in very useful as she spent so much of her time craning over garden fences, spying on the neighbors. The Dursleys had a small son called Dudley and in their opinion there was no finer boy anywhere.

Figure C.1: Samples are shown with tokens highlighted to indicate changes in logits during a given range. Red is improved performance (higher logit output for the true next token) and blue is worse. Sample (a): improvement in bigrams (LM1) such as "te/ll, ab/out, des/ire, mot/ion, eng/aged, strugg/le, etc." Sample (b): improvement in common $n$-grams (LM2) such as "L/in/ux, P/y/th/on, h/on/or/able, S/up/reme, dat/ab/ase, f/ram/ew/ork." Sample (c): development of in-context learning via induction circuits (LM3, LM4), visible in the improved predictions in the word "D/urs/ley" after the first time it appears in the context, as initially observed by (Olsson et al., 2022).

## C.2   Structural development

### C.2.1   Positional embedding

In Figure C.2, we measure the effect of the positional embedding on model performance by comparing the model's performance at particular context positions on a validation set over the course of training against performance on the same validation set but with the positional embedding zero-ablated. The full context length is 1024, and we measure test loss at positions 1, 2, 3, 5, 10, 20, 30, 50, 100, 200, 300, 500, and 1000. In the transition from stage LM1 to LM2, the model begins using the learnable positional embedding to improve performance. The difference between test loss with and without the positional ablation is negligible at all measured positions until the LM1–LM2 boundary.

Structurally, we might predict that the positional embeddings should organize themselves in a particular way: in order to understand relative positions, adjacent positions should be embedded close to each other, and far-away positions should be embedded far apart.

In Figure C.3, we examine the development of the positional embedding itself over time from two angles. The first is to take the embeddings of each position in the context and to run PCA on those embeddings. The result is that as training progresses, the positional embedding PCAs gradually resolve into Lissajous curves, suggesting that the positional embeddings might look like a random walk (Antognini & Sohl-Dickstein, 2018; Shinn, 2023). However, if we look to the explained variance, we see that it grows very large for PC1, reaching 94.2% at training step 6400. This is much higher than we would expect for Brownian motion, where we expect to see about 61% explained variance in PC1 (Antognini & Sohl-Dickstein, 2018).

The second perspective we use is to look at how the magnitudes of positional embeddings over the context length develop. In this case, we observe that the magnitudes seem to have a fairly regular structure. In conjunction with the PCAs and explained variance, we might infer that the positional embeddings look approximately like a (possibly curved) line in $d_{\mathrm{model}} = 256$ dimensional space. A positional embedding

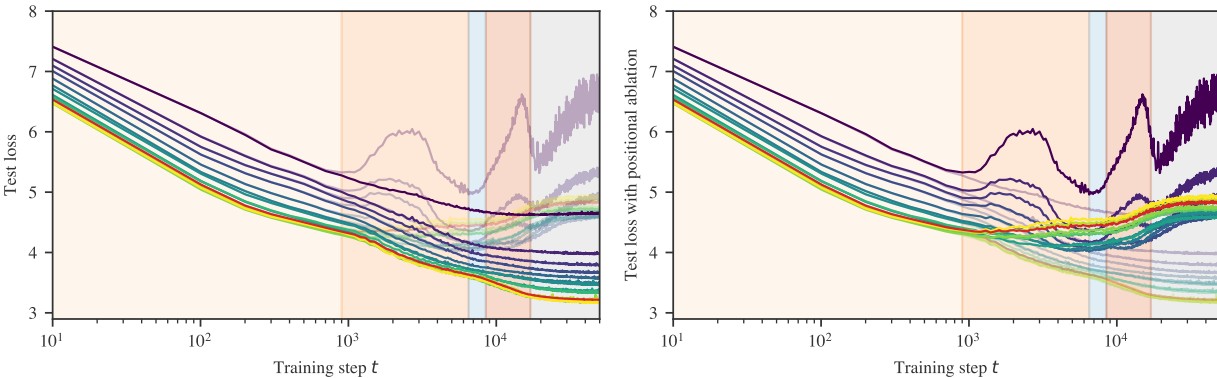

Figure C.2: The model learns to start using the positional encoding in LM2, when the performance starts to worsen when ablating the positional encoding. In both plots, earlier token positions are colored more purple, while later token positions are more yellow, and the overall mean loss is colored in red. Both sets of per-token losses are shown in both graphs for ease of comparison. *Left:* original test loss is emphasized. *Right:* test loss with the positional embedding ablated is emphasized.

organized in this way would make it easier for an attention head to attend to multiple recent tokens, which is necessary if a single head is to learn $n$-grams.

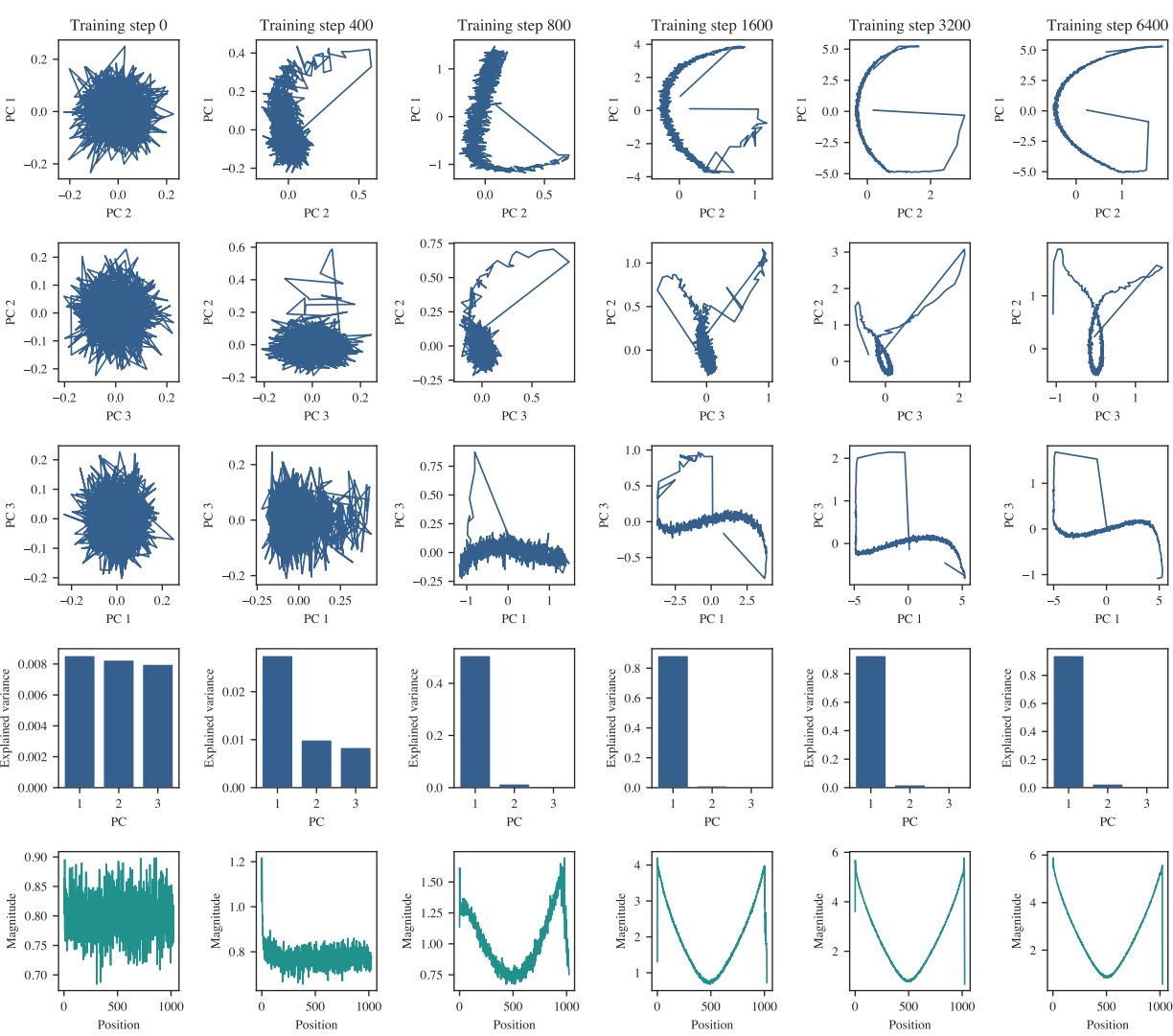

Figure C.3: Columns progress through training time at training steps 0, 400, 800, 1600, 3200, and 6400. The first three rows are plots of the first three principle components of PCA on the positional embedding weights, while the fourth row shows the explained variance for each of the principal components. The fifth row plots the magnitude of the embedding of each position in the context length of 1024.

### C.2.2 Composition scores

Let $W_Q^h, W_K^h, W_V^h$ be the query, key, and value weights of attention head $h$ respectively. There are three types of composition between attention heads in transformer models in Elhage et al. (2021):

- Q-Composition: the query matrix $W_Q^h$ of an attention head reads in a subspace affected by a previous head

- K-Composition: the key matrix $W_K^h$ of an attention head reads in a subspace affected by a previous head

- V-Composition: the value matrix $W_V^h$ of an attention head reads in a subspace affected by a previous head

If $W_O^h$ is the output matrix of an attention head, then $W_{QK}^h = W_Q^h{}^T W_K^h$ and $W_{OV}^h = W_O^h W_V^h$. The composition scores are

$$||MW_{OV}^{h1}||_F / (||M||_F ||W_{OV}^{h_1}||_F) \tag{17}$$

Where $M = W_{QK}^{h_2}{}^T$, $M = W_{QK}^{h_2}$, and $M = W_{OV}^{h_2}$ for Q-, K-, and V-Composition respectively. See Figure C.4 for K-composition scores over time between attention heads in the induction circuits.

### C.2.3 Previous-token matching score

The *previous-token matching score* is a structural measure of induction head attention. It is the attention score given to $[A]$ by an attention head at $[B]$ in the sequence $\ldots[A][B]$ (i.e., how much the head attends to the immediately preceding token).

We compute this score using a synthetic data generating process, generating 10k fixed random sequences with length between 16 and 64. The first token is a special "beginning of string" token, and the remaining tokens are uniformly randomly sampled from other tokens in the vocabulary.

For each sample in this synthetic dataset, we measure the attention score that an attention head gives to the previous token when at the last token in the sequence. These scores are averaged across the dataset to produce the previous-token matching score for that attention head at a given checkpoint. The progression of previous-token matching scores over time can be seen in Figure 4(d).

### C.2.4 Prefix matching score

The *prefix matching score* from Olsson et al. (2022) is defined similarly to the previous-token matching score. Given a sequence $[A][B]\ldots[A]$, the prefix matching score of a particular attention head is how much the attention head attends back to the first instance of $[A]$ when at the second instance of $[A]$.

We compute this score using a synthetic data-generating process. We first generate 10k fixed random sequences of length 128. The first token is always a special "beginning of string" token and the $[A]$ and $[B]$ tokens are selected and placed randomly. One $[A]$ token is placed in the first half of the sequence, the other is placed in the second half, and the $[B]$ token is placed directly after the first $[A]$ token. The remaining tokens are randomly sampled from the tokenizer vocabulary, excluding the $[A]$, $[B]$, and beginning of string tokens.

For each sample in this synthetic dataset, we measure the attention score that each attention head assigns to the earlier instance of $[A]$ from the latter instance of $[A]$. These scores are averaged across the dataset to produce the prefix matching score for that attention head at a given checkpoint. The progression of prefix matching scores over time can be seen in Figure 4(e).

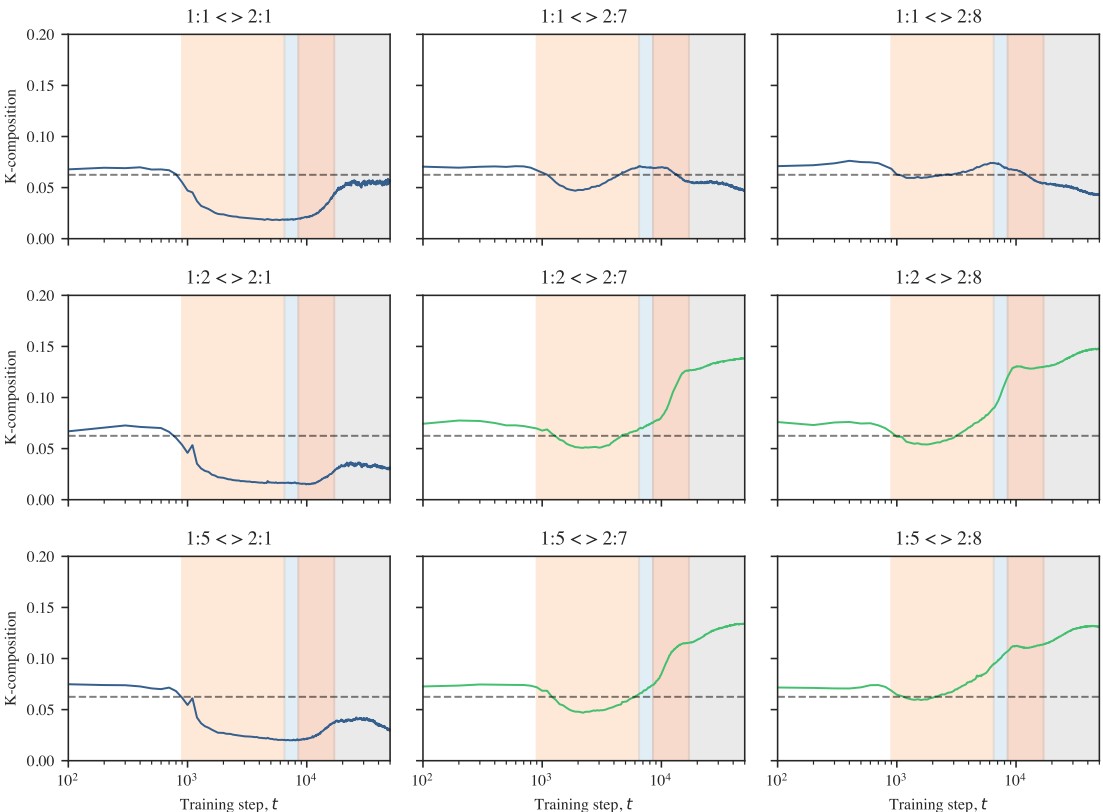

Figure C.4: The K-composition scores (Elhage et al., 2021) between first and second layer attention heads. The $h$th attention head in layer $l$ is indexed by $l : h$. The attention heads that eventually become previous token heads are $h = 2, 5$ in layer 1 (subplot rows 2 and 3), and the attention heads that eventually become induction heads are $h = 7, 8$ in layer 2 (subplot columns 2 and 3). The attention heads $1 : 1$ and $2 : 1$ are included for comparison. The induction heads $2 : 7$ and $2 : 8$ begin K-composing with first layer heads near the start of stage LM2. They continue to compose with the previous token heads in stages LM3 and LM4 (highlighted in green) while their K-composition scores drop with other attention heads in layer 1 in later stages.

# D  Developmental analysis of regression transformers

In this section, we present further evidence on the behavioral (Appendix D.1) and structural (Appendix D.2) development of the transformer in the setting of in-context linear regression.

## D.1  Behavioral development

### D.1.1  Task prior score

In addition to training models on a data distribution in which tasks $\mathbf{t}$ are generated on-the-fly, we examine the setting of Raventós et al. (2023), in which a finite set of $M$ tasks is generated ahead of time, and training samples involve randomly selected tasks from this set.

Figure D.1 depicts (a) the mean square distance between the model's predictions and the zero prediction in addition to (b) the mean square distance between the model's predictions and the "task prior" prediction, using the component-wise averaged $\bar{\mathbf{t}}$ over the set of tasks encountered during training. For all models, the minimum distance to the task prior prediction is lower than the minimum distance to the zero prediction. Hence, we call stage LR1 "learning the task prior" rather than simply learning the zero prediction.

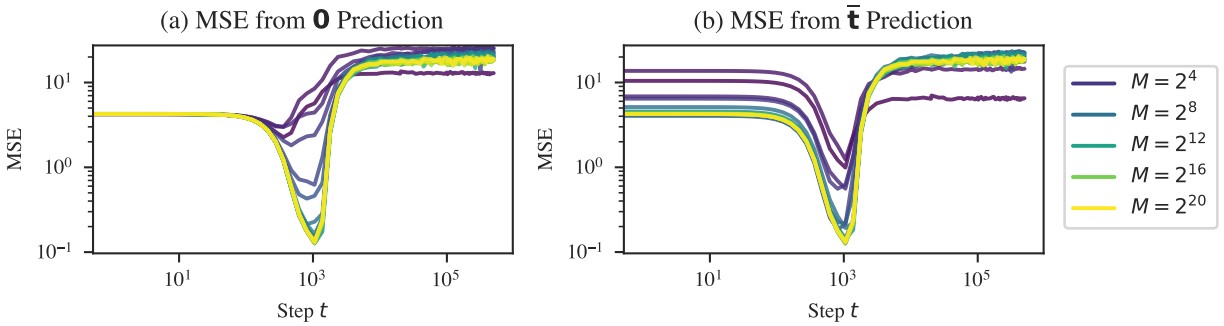

Figure D.1: **Learning the task prior** is universal across models trained on very different data distributions. Each line represents a model trained on a data distribution with a different number of $M$ distinct tasks ("task diversity" in Raventós et al., 2023). In addition to taking a finite $M$, the models depicted here differ from the other models considered in this paper in that the former were trained with a maximum learning rate of 0.01, and the models (inadvertently) lack an output matrix after the multi-head attention layer.

### D.1.2  ICL

We consider two variants of the ICL score: $\text{ICL}_{1:D}$, and $\text{ICL}_{D:K}$.

If the noise term $\sigma^2$ equals zero and both tasks $\mathbf{t}$ and inputs $x_k$ are normalized (i.e., $\mathbf{t} \in S^{D-1}$), then $D-1$ observations of input/output pairs are enough to precisely identify $\mathbf{t}$. Therefore, $\text{ICL}_{1:D}$ measures how successful the model is at initially locating the task. The fact that the tasks and inputs are not normalized changes this only slightly: the task will still sit near $S^{D-1}$ within a shell of vanishing thickness as $D \to \infty$.

Once localized, $\text{ICL}_{D:K}$ measures how successfully the model refines its internal estimate of $\mathbf{t}$ with additional examples, which it can use to reduce the error due to noise.

In terms of implementation, it's not necessary for the model to internally make a distinction between locating and refining its estimate of the task. For example, ridge regression makes no distinction. Still, we find it useful for reasoning about the progression of the model. In particular, we note that early in stage LR2, while the model begins to develop ICL for early tokens, it becomes *worse* at ICL over tokens late in the context. Later, at around 23k steps, $\text{ICL}_{D:K}$ stabilizes, while $\text{ICL}_{1:D}$ continues improving over the entire training run.

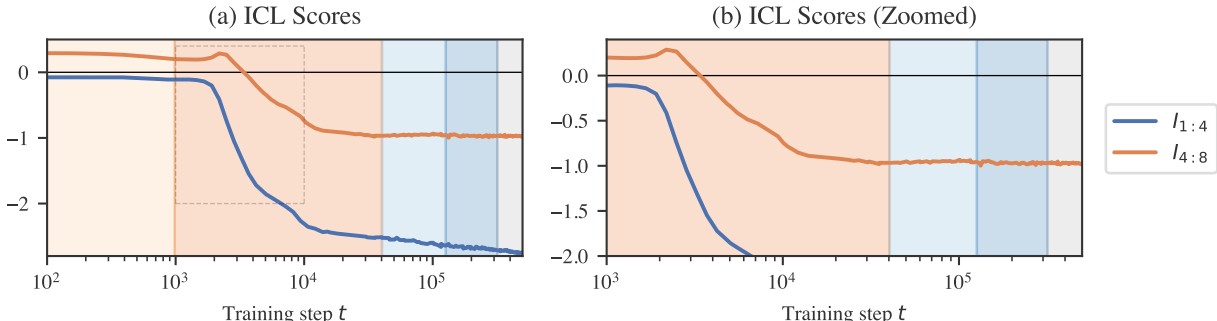

Figure D.2: **ICL scores** for the in-context linear regression model. *Right*: ICL scores between inputs 1 and 4 and inputs 4 and 8 over time. We see that ICL emerges during the first half of LR2. *Left*: Highlighted ICL score curves from the end of LR1 to halfway through LR2. Note that when the model first starts improving on early tokens, it temporarily becomes worse at predicting later tokens. Note also that the model ceases to become better at later tokens as of the second half of LR2, whereas ICL on early tokens continues to improve throughout training.

### D.1.3 OOD generalization

To further investigate behavior in stages LR2 and LR3, we probe the model on data sampled from different distributions than encountered during training.[1] We evaluate behavior on two families of perturbations: "OOD inputs" $x_k$, sampled according to a different scale

$$x_k \sim \mathcal{N}(0, gI_D), \tag{18}$$

for some gain parameter $g$, and "OOD tasks"

$$\mathbf{t} \sim \mathcal{N}(0, gI_D). \tag{19}$$

Note that these inputs and tasks are not out-of-distribution in the sense of coming from a distribution with a different support than the training distribution. However, the samples drawn from these "extreme" distributions are exponentially suppressed by the original training distribution. Figure D.3 plots the normalized MSE for these two distributions over training time.

Between $t = 1k$ and $t = 4k$ the model's outputs rapidly *diminish* in scale for out-of-distribution samples, both for $g > 1$ and $g < 1$, especially for out-of-distribution *inputs*. While the model is moving away from predicting with the task prior for in-distribution samples, it moves closer to predicting with the task prior for-in-distribution samples.

Between $t = 4k$ and $t = 23k$, the model recovers on moderately out-of-distribution inputs $g < 10^{1.5}$ with performance remaining close to constant beyond this range. Past this stage, performance improves constantly for out-of-distribution tasks.

For out-of-distribution inputs, performance eventually worsens for some ranges of $g$. Between $t = 23k$ and $t = 80k$ the model further approaches the task prior prediction for extreme out-of-distribution inputs $g > 10^{1.5}$ . Subsequently, between $t = 75k$ and $t = 130k$ the model moves away from the task prior prediction for extreme inputs, and performance deteriorates for inputs with $g > 10^{0.5}$. As of LR5, performance is roughly constant.

---

[1]Cf. Raventós et al. (2023) evaluating models trained on a set of discrete tasks on the "true" distribution consisting of novel tasks.

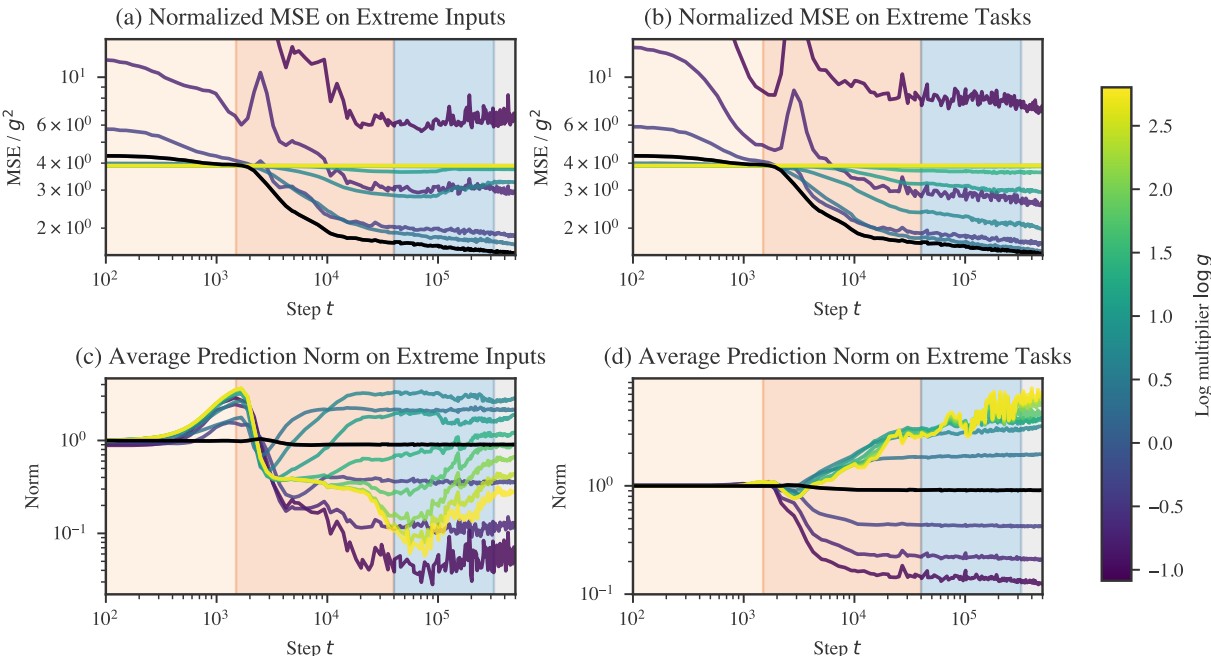

Figure D.3: Performance on extreme inputs over time may reveal additional substages in LR2 and in LR3. *Left*: The model first becomes better, then worsens at ICL on inputs sampled from $\mathcal{N}(0, gI_D)$ for large $g$. *Right*: The model continues to improve on ICL at tasks sampled from $\mathcal{N}(0, gI_D)$. *Top*: Normalized loss (divided by $g^2$) over time for OOD inputs and tasks. *Bottom*: Average $|\hat{y}|$ over time for OOD inputs and tasks.

## D.2 Structural development

### D.2.1 Embedding

The embedding matrix $W_E$ is a linear transformation from $\mathbb{R}^{D+1} \to \mathbb{R}^{d_{\mathrm{embed}}}$. Plotting the $D+1$ singular values of this matrix, we notice that the embedding partially loses one of its components starting at the end of LR2 (Figure D.4a).

The input "tokens" $x_k$ span a $D$-dimensional subspace of the $(D+1)$-dimensional "token space." The target tokens $y_k$ span an orthogonal 1-dimensional subspace. The collapse of one of the embedding matrix's singular values means that the model learns to redundantly encode the inputs and targets in the same $D$-dimensional subspace of the space of residual stream activations. The almost order of magnitude separation in the magnitudes of the square singular value means that the $(D+1)^{\mathrm{th}}$ component of the token embedding explains only 2.9% of the variance in activations of the residual stream immediately after the embedding, whereas the dominant components explain roughly 24% each.

**Contributions to degeneracy** Given a linear transformation $T_1 : \mathbb{R}^{D_1} \to \mathbb{R}^{D_2}$ followed by another linear transformation $T_2 : \mathbb{R}^{D_2} \to \mathbb{R}^{D_3}$, reducing the rank of $T_1$ from $r$ to $r' < r$ renders $D_3(r - r')$ components of the second transformation irrelevant. This would mean a *decrease* in the learning coefficient of $D_3(r-r')/2$ (a decrease in the *effective dimensionality* of $d$ leads to a decrease in the LLC of $d/2$[2]). In the actual model, we don't see an exact decrease in the rank, and a layer normalization sits between the linear transformation of the embedding and the linear transformations of each transformer block and unembedding. It is unclear what the precise relation between structure and degeneracy is in this case (Appendix D.2.6). Still, suggestively, the onset of embedding collapse coincides with a decrease in the rate of increase of $\hat{\lambda}(w_t)$.

---

[2]Note that this is not the only possible way for the LLC to decrease. Changing the local loss landscape from quadratic to quartic or some higher power would also lower the LLC, by a fractional amount.

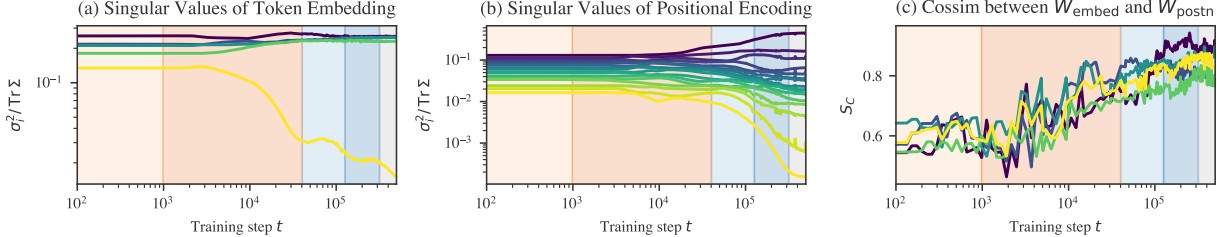

Figure D.4: *Left*: The embedding partially "collapses" during the second half of LR2. At the start of stage LR2, the minimum singular values explains only 3% of the variance in residual stream activations due to the sample. By the end of training, it explains half that. *Middle*: The positional encoding goes through a similar shift during LR3 (that begins earlier during LR2). *Right*: The cosine similarity between the 5 rows of $W_\text{embed}$ and the projection of those rows onto the subspace spanned by $W_\text{unembed}$ shows that the model learns to write to the same write tokens and positional information to the same subspace.

### D.2.2 Positional encoding

The positional encoding goes through a similar collapse to the unembedding starting during the second part of LR2 and continuing into LR3 (Figure D.4b). Additionally, throughout these stages, the subspace spanned by the embedding becomes more aligned with the subspace spanned by the positional encoding (Figure D.4c).

**Contributions to degeneracy** For the same reason as with the token embedding, a decrease in the dimensionality of the subspace occupied by activations reduces the effective number of dimensions and thus the learning coefficient. This occurs both as the positional encoding's effective dimensionality decreases (vanishing singular values, Figure D.4b) and as the token embedding subspace and positional embedding subspace align (increasing cosine similarity, Figure D.4b).

### D.2.3 Attention collapse

Over the course of training, we observe that some attention heads learn to attend *solely* (soft attention becomes hard attention) and *consistently* to certain positions (the attention pattern becomes content-independent). We call this phenomenon *attention collapse* in parallel with the other observed forms of collapse. Not only does this potentially contribute to a decrease in the LLC, but it also makes the attention heads identifiable: we find a self-attention head, previous-attention heads, previous-$x$-attention heads, and previous-$y$-attention heads.

**$x$-attention vs. $y$-attention** For convenience we separate each attention head in two: one part for the $x$-tokens, and the other for the $y$-tokens.

**Attention entropy score** To quantify attention hardness, we use the *attention entropy score* (Ghader & Monz, 2017; Vig & Belinkov, 2019). Given the attention pattern $\alpha_{k,k'}^{(b,h)}$ for how much token $k$ in head $h$ in block $b$ attends back to token $k'$, its attention entropy score $H_k^{(b,h)}$ is the Shannon entropy over preceding indices $k' < k$,

$$H_k^{(b,h)} = -\sum_{k' \leq k} \alpha_{k,k'}^{(b,h)} \log_2 \alpha_{k,k'}^{(b,h)}. \tag{20}$$

From this, we compute the normalized entropy $\hat{H}_k^{(b,k)}$, which divides the attention entropy by the maximum entropy for the given context length,

$$\hat{H}_k^{(b,h)} = \frac{H_k^{(b,h)}}{\log_2(k)}. \tag{21}$$

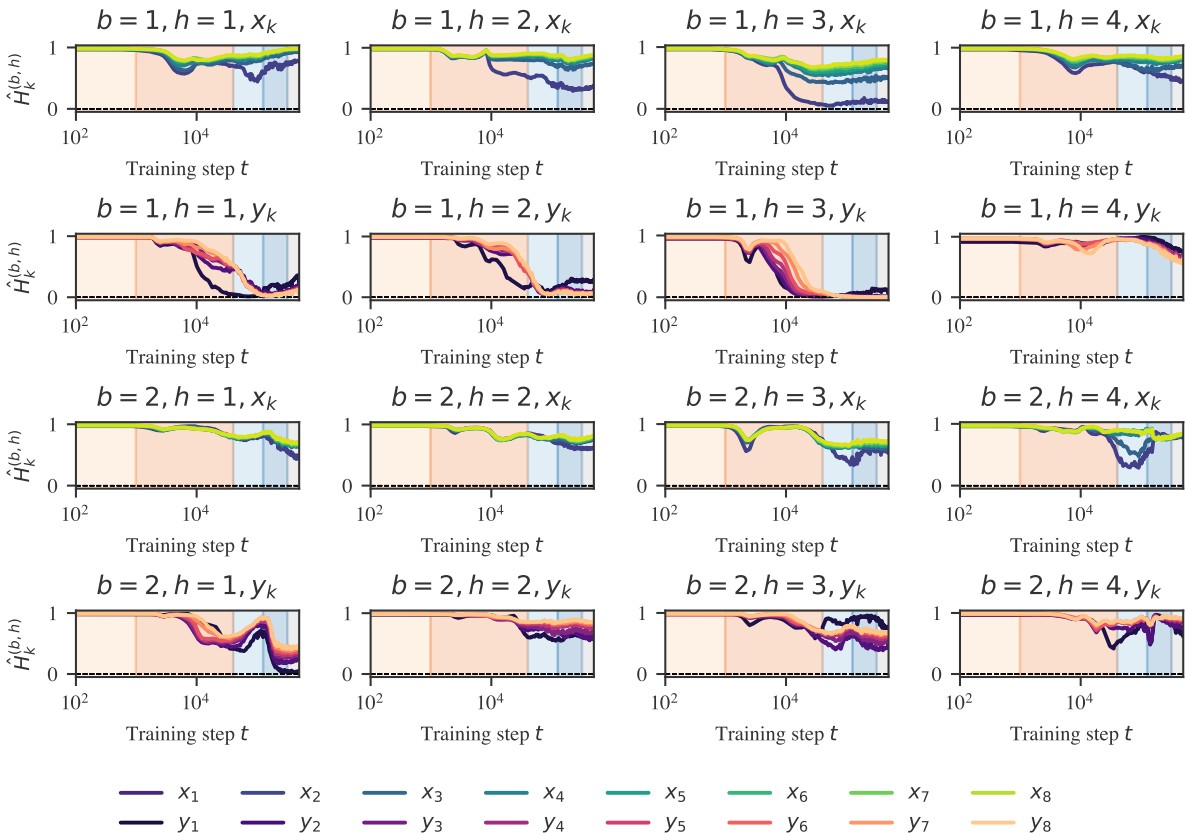

Figure D.5: **Attention hardening** as measured by the normalized attention entropy score (Appendix D.2.3). Block 1 heads 1y/3y and block 2 head 1y harden over training. In combination with the fact that these attention heads become less variable (Figure D.6), this may contribute to a decrease in the LLC (discussed in Appendix D.2.3) The x-components of the attention heads remain much softer over the entire training run.

This accounts for the entropy being calculated over different numbers of tokens and is displayed in Figure D.5. Notably, the identified stages line up closely to stages of these attention entropy curves.

**Constant attention**   Accomplishing constant attention requires the presence of biases in the query and key transformations, or if there is no bias (as is the case for the models we investigated), requires attending to the positional embedding. With the Shortformer-style positional encoding used for the language models (Appendix F.1.1), this is straightforward: the positional information is injected directly into the key and weight matrices. With the in-context linear regression models, where the positional embedding is added to the residual stream activations, this is less straightforward: achieving constant attention requires separating residual stream activations into orthogonal positional- and input-dependent subspaces, then reading from the former with the query and key weight matrices.

**Attention variability score**   To quantify how constant the attention pattern is, we use measure *attention variability* (Vig & Belinkov, 2019),

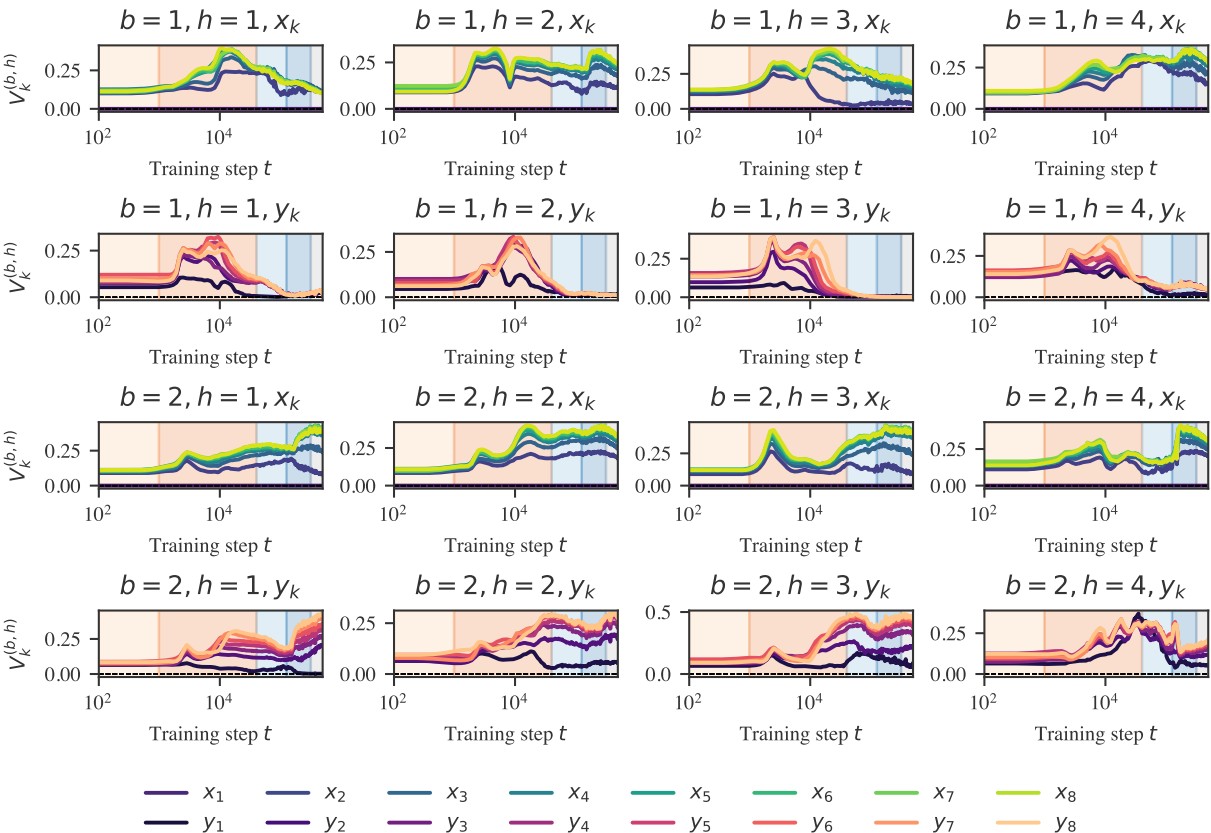

Figure D.6: **Attention variability** over time. The heads that develop hard attention in Figure D.5 (block 1 heads 1y, 3y, and 4y) also become less variable over time.

$$V_k^{(b,h)} = \frac{\sum_{i=1}^n \sum_{k' \le k} \left| \alpha_{k,k'}^{(b,h)}(S_K^{(i)}) - \bar{\alpha}_{k,k'}^{(b,h)} \right|}{2n \sum_{k' \le k} \bar{\alpha}_{k,k'}^{(b,h)}}, \tag{22}$$

where the division by 2 ensures the variability lies in the range $[0, 1]$. This is displayed in Figure D.6. These reveal that though attention hardness and variability are independent axes of differentiation, empirically, we observe that hard attention is correlated with low variability.

**Self-attention score**   Self-attention is measured by the average amount a token $k$ attends to itself, $\alpha_{k,k}^{(b,h)}$.

**Previous-token attention score**   Previous-token attention is measured the same as in the language model setting (Appendix C.2) with one difference: we compute the previous-token score not over a synthetic dataset but over a validation batch.

$x$**-attention score**   The total amount attended to inputs $x_k$, that is $\alpha_{k,x}^{(b,h)} = \sum_{k'=1}^K \alpha_{k,2k}^{(b,h)}$.

$y$**-attention score**   Defined analogously $\alpha_{k,x}^{(b,h)} = \sum_{k'=1}^K \alpha_{k,2k+1}^{(b,h)}$.

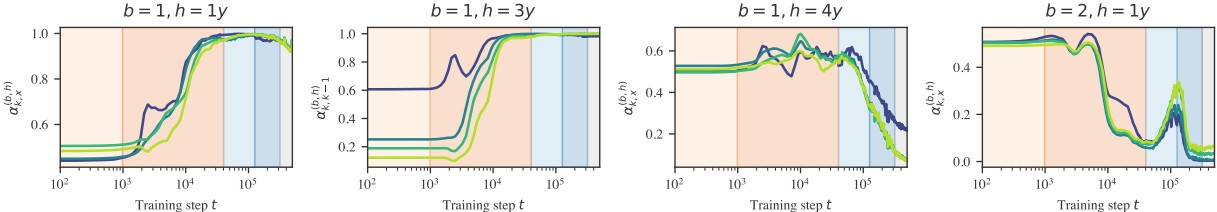

Figure D.7: Collection of attention heads identified by their consistent and recognizable attention patterns. *Left to right*: previous-$x$s head, previous-token head, previous-$y$s head, previous-$y$s head

**Classifying attention heads**  Several attention heads are easy to identify by virtue of being both concentrated and consistent. These are depicted in Figure D.7 and include: (B1H3y) previous-token heads (also present in the language model case), (B1H1y) previous-x, and (B1H4x, B2H1y) previous-y heads. Other training runs also include self-attention heads.

**Contributions to degeneracy**  Suppose an attention head $h$ in block $b$ has the following constant attention pattern (after the softmax) $A^{(b,h)} = \sum_i \delta_{l(i)\,i}$. That is, for each token $i$, that attention head attends solely to a single earlier token $l(i) \leq i$ and no others. Restricting to single-head attention (the argument generalizes straightforwardly), the final contribution of this attention head to the residual stream is the following (Phuong & Hutter, 2022):

$$O = W_O \cdot (V \cdot A) \tag{23}$$

where $A \in \mathbb{R}^{\ell_z} \times \mathbb{R}^{\ell_x}$ is the attention pattern, $V \in \mathbb{R}^{d_{\text{out}}} \times \mathbb{R}^{\ell_z}$ is the value matrix, and $W_O \in \mathbb{R}^{d_z} \times \mathbb{R}^{\ell_z}$ is the matrix of residual stream activations, and $V \in \mathbb{R}^{d_{\text{out}}} \times \mathbb{R}^{\ell_z}$ is the value matrix. The result of this operation is subsequently multiplied by the output matrix and then added back into the residual stream. Plugging in the hard and constant attention pattern, writing out the matrix multiplication, and filling in the definition of $A$ we get

$$O_{ij} = \sum_k (W_O)_{ik} V_{kl(j)} \delta_{l(j)j}. \tag{24}$$

For each column in $A$, the hard attention picks out a single element of $V$ at column $l(j)$ for each row $k$. Now suppose that there is a token $l'$ that receives no attention from any position $j$. That is, there exists no $j$ such that $l' = l(j)$. Then, there is a column $l'$ in $V$ which does not contribute to the result of $V \cdot A$, and, in turn, a column $l'$ in $W_O$, which does not contribute to the output of the head. As discussed for the embedding and layer norm, this decrease in effective dimensionality leads to a decrease in the learning coefficient.

Note that this argument does not hold for all hard and constant attention patterns. It holds solely for attention patterns that consistently ignore some earlier token across all positions, such as the previous-$x$ and previous-$y$ heads, but not the self-attention and previous-token heads. As discussed in Appendix D.2.6, it remains unclear what exactly the threshold for "ignoring" a token should be before it contributes to degeneracy and whether any of the heads we examine actually meet this threshold.

### D.2.4  Unembedding collapse

The unembedding block consists of a layer normalization layer $\text{LN}(z)$ followed by a linear transformation $W_U z + b_U$ and finally a projection $\pi_y$ to extract the $y$-component. Given the 64-dimensional vector of activations $z$ in the residual stream right before the unembedding (for a specific token), the full unembedding operation is:

$$\pi_y \left[ W_U \left( \frac{z - \mathbb{E}[z]}{\sqrt{\mathbb{V}[z] + \epsilon}} \odot \gamma + \beta \right) + b_U \right]$$

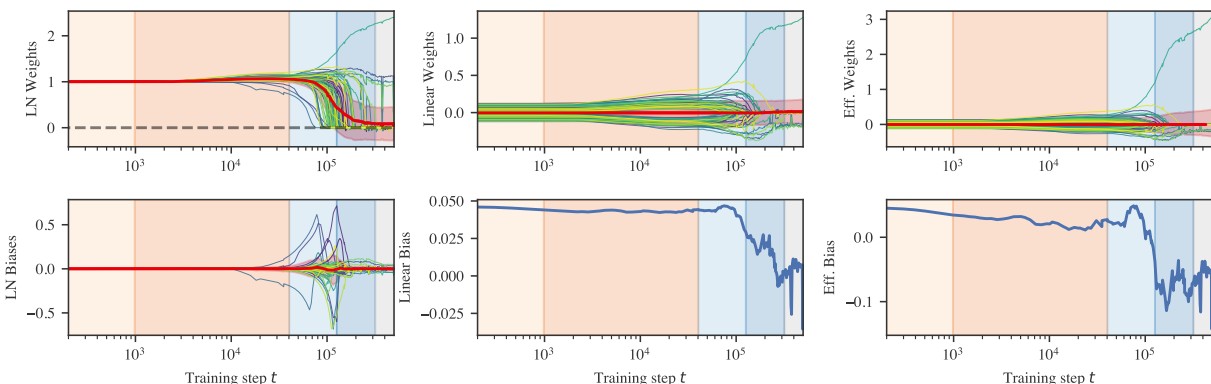

Figure D.8: **Unembedding weights over time** for the RT1 transformer undergo a "collapse" that begins towards the end of LR2. When these weights reach zero in LR3 and LR4, it may contribute to the observed decrease in the LLC. *Top*: Weights over time. The outlier in the positive direction is the weight for the $y$-token output. *Bottom*: Biases over time. *Left*: Unembedding layer normalization weights over time. *Middle*: Unembedding linear weights over time (restricted to $y$-subspace). *Right*: Effective unembedding weights over time (obtained by element-wise multiplication of preceding columns, and focusing on the bias for only the $y$-token.

where $\odot$ denotes element-wise multiplication of two vectors and $\gamma, \beta$ are the layer normalization weights and biases respectively.

**Effective unembedding weights and biases**   Moving terms around, we can represent this as

$$\left((W_U)_{[0,:]} \odot \gamma\right) \left(\frac{z - \mathbb{E}[z]}{\sqrt{\mathbb{V}[z] + \epsilon}}\right) + \left((W_U)_{[0,:]}\beta\right) + (b_U)_{[0]}$$

where we order the outputs so that the $y$-token corresponds to the 0th row. Because we are reading out a single $y$ component, we can express the unembedding transformation in terms of "effective" unembedding weights and biases

$$\tilde{W}_U = (W_U)_{[0,:]} \odot \gamma,$$
$$\tilde{b}_U = \left((W_U)_{[0,:]}\beta\right) + (b_U)_{[0]}.$$

**Unembedding weights over time**   In Figure D.8, we plot $(\gamma, \beta)$, $((W_U)_{[0,:]}, (b_U)_{[0]})$, and $(\tilde{W}_U, \tilde{b}_U)$ as a function of training steps, along with the mean weight over time. These are 64- and 1-dimensional vectors, so we can display the entire set of components. During stage LR3 the majority of weights $\beta$ and $W_U$ "collapse" to zero. Additionally, the layer normalization biases temporarily experience a large increase in variance before returning to small values. Despite this, the mean of the linear weights, layer normalization biases, and effective weights remains remarkably constant and close to zero throughout the entire process.

**Contributions to degeneracy**   Suppose that $D$ of the layer normalization weights have vanished, say $\gamma_i = 0$ for $1 \leq i \leq D$. Then the corresponding columns of $W_U$ only contribute to the unembedding via their product $(W_U)_{[:,1:D]}\beta_{[1:D]}$ with the first $D$ rows of $\beta$. This creates a typical form of degeneracy studied in SLT and found, for example, in deep linear networks, where we can change the weights to $(W_U)_{[:,1:D]}A, A^{-1}\beta_{[1:D]}$ for any invertible $D \times D$ matrix $A$ without changing the function computed by the network. If in addition the $\beta_i$ vanish for $1 \leq i \leq D$ then the entries of $(W_U)_{[:,1:D]}$ are completely unconstrained, creating further degeneracy.

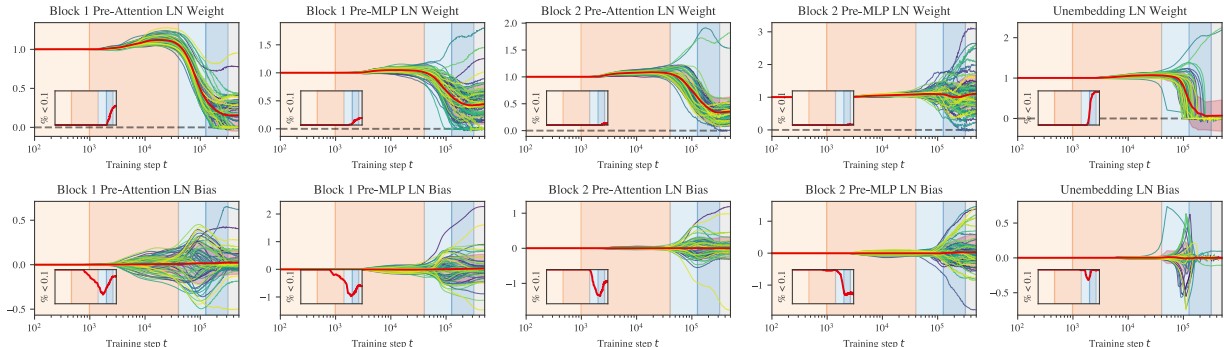

Figure D.9: **Layer norm weights over time**. *Top*: After LR3, the layer normalization collapse expands from the unembedding to earlier layers, most notably in the first pre-attention layer norm. This occurs without explicit regularization and may contribute to the concurrent decrease in LLC. *Bottom*: During layer normalization collapse, the variance of layer normalization biases increases drastically while the mean of the biases remains relatively constant. *Inset*: Plotting the fraction of weights or biases whose magnitude is less than 0.1 over time reveals that the collapse is more measured for intermediate layer norms: weights shrink to small values but not extremely close to zero as in the unembedding and first attention layer.

### D.2.5 Layer normalization collapse

The "collapse" in layer normalization weights is not unique to the unembedding. As depicted in Figure D.9, this behavior occurs in all layer norms except for the second MLP. The biases also remain centered close to zero even as the variance in biases grows much larger. Unlike in the unembedding, these layers begin to change earlier (starting halfway through LR2).

What is most striking about the layer normalization collapse is that it occurs without any explicit regularization (neither weight decay nor dropout). As such, it demonstrates a clear example of *implicit regularization*, i.e., inductive biases in the optimizer or model that favor simpler solutions.

**Contributions to degeneracy**   In the previous section, we describe how layer norm collapse in the unembedding is linked to an increase in degeneracy because it ensures that parameters in the subsequent linear layer become irrelevant. The same is true for layer norm which precedes the attention and MLP blocks.

### D.2.6 Degeneracy and development

In the previous subsections, we provide a set of theoretical arguments for how embedding collapse (Appendix D.2.1), layer normalization collapse (Appendix D.2.5), and attention collapse (Appendix D.2.3) can lead to an increase in degeneracy, even while leaving the implemented function unchanged.

The free energy formula tells us that, for two different solutions (sets of weights) with the same loss, the Bayesian posterior will asymptotically prefer the model that has the lower learning coefficient (i.e., higher degeneracy). This suggests that these different forms of collapse may be driven by a bias towards higher degeneracy, as captured in the free energy formula. However, in idealized Bayesian inference, we do not expect the posterior to concentrated around the neighborhood of an equal-loss-but-higher-degeneracy local minimum to begin with. That this kind of transition arises in practice might arise from one of the various differences between Bayesian inference and gradient-based training.

Actually establishing a causal link between increasing degeneracy and structure development is beyond the scope of this paper. For one, the theoretical arguments hinge on the collapse being *complete*, that is, the components that go to zero must become *exactly* zero in the limit, where we take the number of samples to compute the loss to infinity. In practice, we expect there to be some threshold $\epsilon$ below which we can treat weights as effectively zero. Second, even if these explanations are correct, we do not know that they account for all of the empirically observed decrease in the LLC during these stages. There may be other drivers

we missed. Finally, establishing a causal link requires theoretical progress in relating the Bayesian learning process to the SGD learning process. The arguments are suggestive, but currently only a source of intuition for how structure and degeneracy can be related, and a starting point for future research.

# E   One-layer language model experiments

We also trained and ran some experiments on a one-layer language model (see Appendix F.1.1 for details). We aggregate results for the one-layer language model here, mirroring the experiments for the two-layer language model where possible. The early development of the one-layer model has many parallels with the two-layer model. At a single stage boundary, just as it occurs in the two-layer model, the one-layer model minimizes its bigram score (see Appendix C.1.1), begins utilizing the positional embedding to noticeably improve performance (see Appendix C.2.1), and starts making sudden improvements to the same $n$-gram scores (see Appendix C.1.2). Remarkably this occurs at the same checkpoint as in the 2-layer model (at 900 training steps).

One key difference, however, is that this occurs at the *second* stage boundary as discerned by the plateaus of the LLC estimation. We did not closely investigate why the LLC estimation appears to drop between steps 400 and 900 in this model. As a result though, we do observe an interesting qualitative similarity to the drop in LLC in stage LM3 of the two-layer model, that this drop precedes a noticeable bump in the loss function.

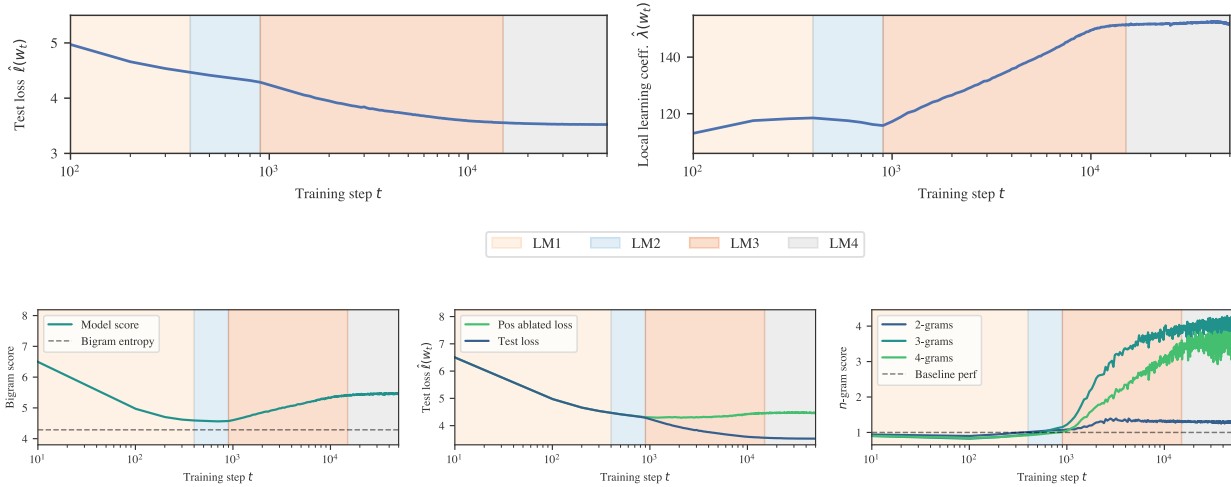

Figure E.1: We train a one-layer transformer model in the language setting to compare with the two-layer model. The development of certain behavioral and structural metrics over time closely mirrors the development of the same metrics in the early stages of the two-layer language model. *Top:* test loss and LLC estimations over time for the one-layer attention-only transformer, compare with Figure 1a. *Bottom:* bigram score, test loss with positional embedding ablated, and $n$-gram scores for the one-layer attention-only transformer, compare with Figure 4(a,b,c).

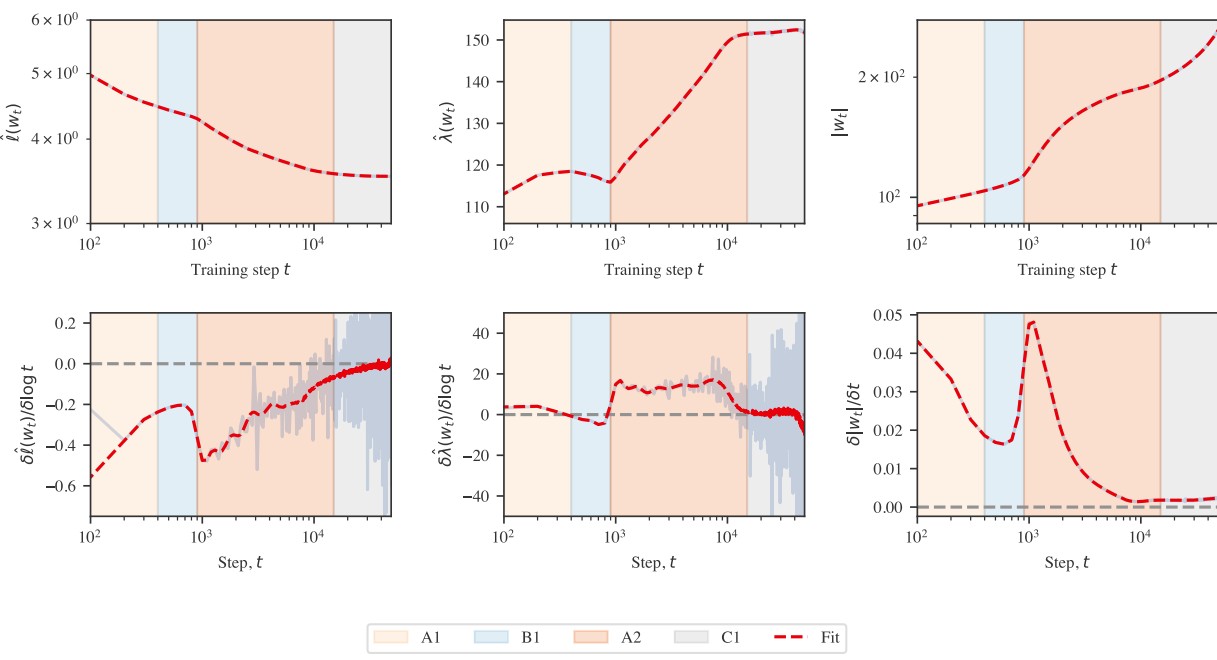

Figure E.2: A more detailed version of Figure E.1 for the one-layer language model. *Top*: Loss, LLC, and weight norm, along with an overlaid Gaussian process fit to these curves (red dotted lines). *Bottom*: Associated slopes, both numerically estimated finite differences (transparent blue) and of the Gaussian process (red dotted lined).

# F    Transformer training experiment details

## F.1    Language models

### F.1.1    Architecture

The language model architectures we consider are one- and two-layer attention-only transformers. They have a context length of 1024, a residual stream dimension of $d_{model} = 256$, $H = 8$ attention heads per layer, and include layer normalization layers. We also used a learnable Shortformer positional embedding (Press et al., 2021). The resulting models have a total of $d = 3,091,336$ parameters for $L = 1$ and $d = 3,355,016$ parameters for $L = 2$. We used an implementation provided by TransformerLens (Nanda & Bloom, 2022).

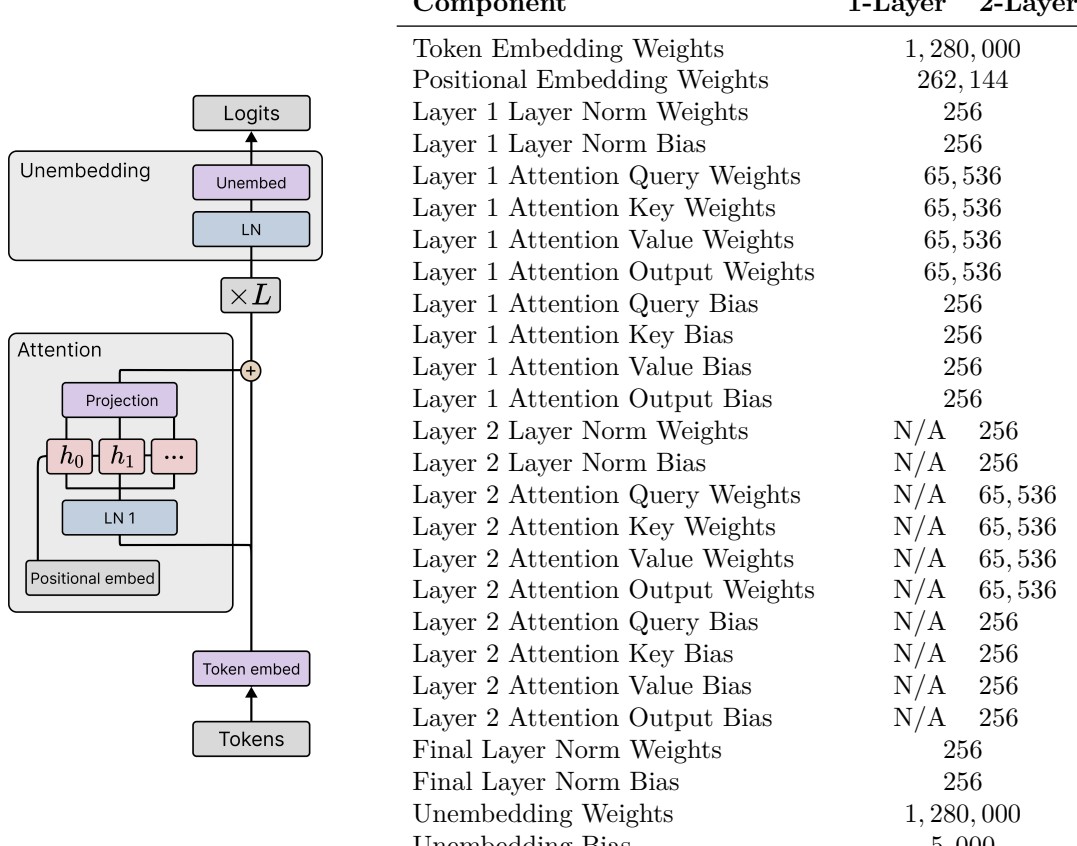

| Component | 1-Layer | 2-Layer |
|---|---|---|
| Token Embedding Weights | 1,280,000 | |
| Positional Embedding Weights | 262,144 | |
| Layer 1 Layer Norm Weights | 256 | |
| Layer 1 Layer Norm Bias | 256 | |
| Layer 1 Attention Query Weights | 65,536 | |
| Layer 1 Attention Key Weights | 65,536 | |
| Layer 1 Attention Value Weights | 65,536 | |
| Layer 1 Attention Output Weights | 65,536 | |
| Layer 1 Attention Query Bias | 256 | |
| Layer 1 Attention Key Bias | 256 | |
| Layer 1 Attention Value Bias | 256 | |
| Layer 1 Attention Output Bias | 256 | |
| Layer 2 Layer Norm Weights | N/A | 256 |
| Layer 2 Layer Norm Bias | N/A | 256 |
| Layer 2 Attention Query Weights | N/A | 65,536 |
| Layer 2 Attention Key Weights | N/A | 65,536 |
| Layer 2 Attention Value Weights | N/A | 65,536 |
| Layer 2 Attention Output Weights | N/A | 65,536 |
| Layer 2 Attention Query Bias | N/A | 256 |
| Layer 2 Attention Key Bias | N/A | 256 |
| Layer 2 Attention Value Bias | N/A | 256 |
| Layer 2 Attention Output Bias | N/A | 256 |
| Final Layer Norm Weights | 256 | |
| Final Layer Norm Bias | 256 | |
| Unembedding Weights | 1,280,000 | |
| Unembedding Bias | 5,000 | |

Figure F.1: **Attention-only transformers** with Shortformer position-infused attention and pre-layer norm. The one-layer model has a total of 3,091,336 trainable parameters, while the two-layer model has 3,355,016.

### F.1.2    Tokenization

For tokenization, we used a truncated variant of the GPT-2 tokenizer that cut the original vocabulary of 50,000 tokens down to 5,000 (Eldan & Li, 2023) to reduce the size of the model. We think this may contribute to the prominence of the the plateau at the end of LM1: the frequency of bigram statistics depends on your choice of tokens, and a larger tokenizer leads to bigrams that are individually much less frequent.

### F.1.3    Training

The models are trained on a single epoch over 50,000 steps on ~5 billion tokens using a resampled subset of the Pile (Gao et al., 2020; Xie et al., 2023) using a batch size of 100. A snapshot was saved every 10 steps

for a total of 5000 checkpoints, though a majority of analysis used checkpoints every 100 steps. The training time was around 6 GPU hours per model on an A100. Additional seeds were trained on v4 TPUs at around 1.5 TPU hours per model.

Training was conducted on the first 10 million lines of the DSIR-filtered Pile (Xie et al., 2023; Gao et al., 2020) but did not exhaust all 10 million lines. The model was subject to weight decay regularization, without the application of dropout. We did not employ a learning rate scheduler throughout the training process.

Table 3: Summary of hyperparameters and their values for transformer language model training experiments.

| Hyperparameter | Category | Description/Notes | Value |
|---|---|---|---|
| $n$ | Data | # of training samples | $5,000,000$ |
| $T$ | Data | # of training steps | $50,000$ |
| $N_{\text{test}}$ | Data | # of test samples | 512 |
| Tokenizer Type | Data | Type of Tokenizer | Truncated GPT-2 Tokenizer |
| $D$ | Data | Vocabulary size | 5,000 |
| $K$ | Data | Context size | 1,024 |
| $L$ | Model | # of layers in the model | 2 |
| $H$ | Model | # of heads per layer | 8 |
| $d_{\text{mlp}}$ | Model | MLP hidden layer size | N/A |
| $d_{\text{embed}}$ | Model | Embedding size | 256 |
| $d_{\text{head}}$ | Model | Head size | 32 |
| seed | Model | Model initialization | 1 |
| m | Training | Batch Size | 100 |
| Optimizer Type | Optimizer | Type of optimizer | AdamW |
| $\eta$ | Optimizer | Learning rate | 0.001 |
| $\lambda_{\text{wd}}$ | Optimizer | Weight Decay | 0.05 |
| $\beta_{1,2}$ | Optimizer | Betas | $(0.9, 0.999)$ |

## F.2 In-context linear regression transformers

### F.2.1 Architecture

In the following $L$ refers to the number of layers (blocks) in the transformer, $H$ is the number of heads in each layer, $D$ is the dimension of inputs $x \in \mathbb{R}^D$ and $K$ is the number of $(x, y)$ pairs provided to the Transformer in-context.

The architecture is a pre-layer-norm decoder-only transformer modeled after NanoGPT (Karpathy, 2022; see also Phuong & Hutter, 2022) with a learnable positional embedding. For the models discussed in the main body, we consider $L = 2$, $H = 4$ transformers (with $d = 51,717$ parameters), i.e., two transformer blocks with four attention heads each.

### F.2.2 Tokenization

To run contexts $S_K$ through the above model requires an initial encoding or "tokenization step" and final "projection step." The context is encoded as a sequence of "tokens" $T_k$ as follows:

$$T_k = \left( \begin{pmatrix} 0 \\ | \\ x_1 \\ | \end{pmatrix}, \begin{pmatrix} y_1 \\ 0 \\ \vdots \\ 0 \end{pmatrix}, \cdots \begin{pmatrix} 0 \\ | \\ x_k \\ | \end{pmatrix}, \begin{pmatrix} y_k \\ 0 \\ \vdots \\ 0 \end{pmatrix} \right).$$

Through the main text, we write $f_w(S_k)$ for $f_w(T_k)$. Note that this tokenization includes the final $y_k$ token even though this receives no training signal. For this reason, we omit this token from the attention entropy and variability plots (Figures D.5 and D.6).

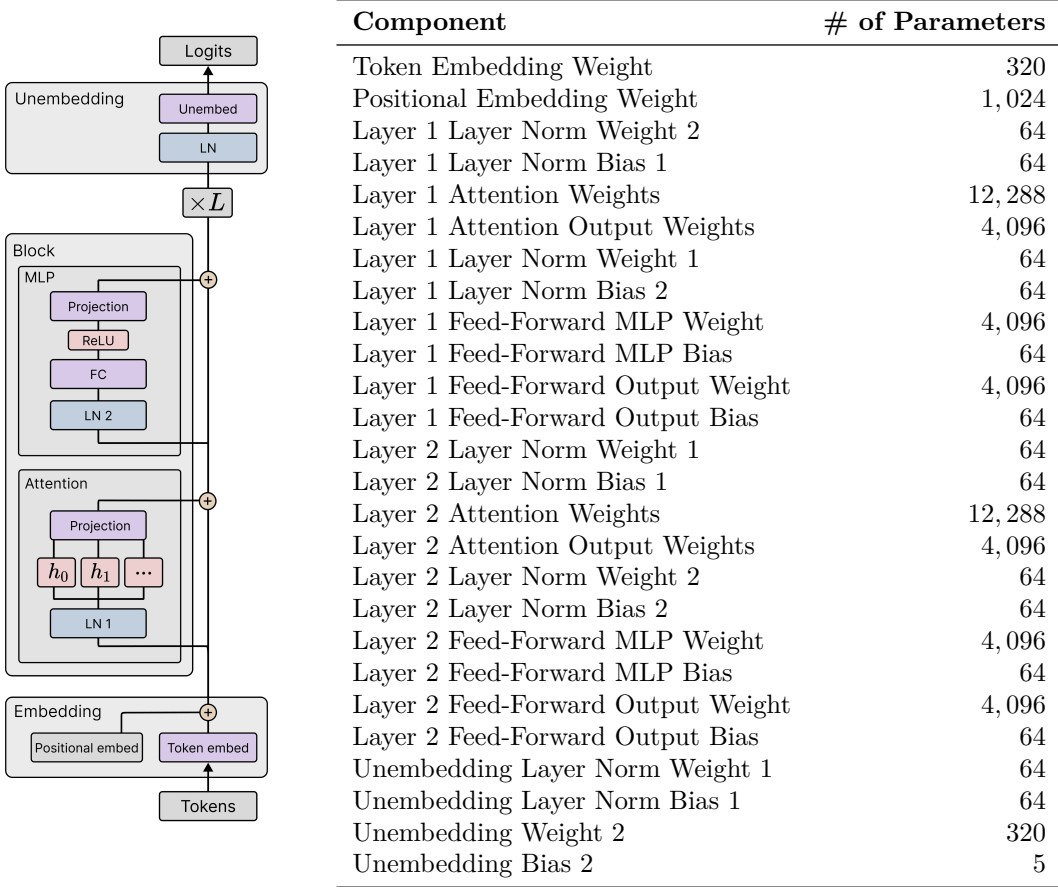

| Component | # of Parameters |
|---|---|
| Token Embedding Weight | 320 |
| Positional Embedding Weight | 1,024 |
| Layer 1 Layer Norm Weight 2 | 64 |
| Layer 1 Layer Norm Bias 1 | 64 |
| Layer 1 Attention Weights | 12,288 |
| Layer 1 Attention Output Weights | 4,096 |
| Layer 1 Layer Norm Weight 1 | 64 |
| Layer 1 Layer Norm Bias 2 | 64 |
| Layer 1 Feed-Forward MLP Weight | 4,096 |
| Layer 1 Feed-Forward MLP Bias | 64 |
| Layer 1 Feed-Forward Output Weight | 4,096 |
| Layer 1 Feed-Forward Output Bias | 64 |
| Layer 2 Layer Norm Weight 1 | 64 |
| Layer 2 Layer Norm Bias 1 | 64 |
| Layer 2 Attention Weights | 12,288 |
| Layer 2 Attention Output Weights | 4,096 |
| Layer 2 Layer Norm Weight 2 | 64 |
| Layer 2 Layer Norm Bias 2 | 64 |
| Layer 2 Feed-Forward MLP Weight | 4,096 |
| Layer 2 Feed-Forward MLP Bias | 64 |
| Layer 2 Feed-Forward Output Weight | 4,096 |
| Layer 2 Feed-Forward Output Bias | 64 |
| Unembedding Layer Norm Weight 1 | 64 |
| Unembedding Layer Norm Bias 1 | 64 |
| Unembedding Weight 2 | 320 |
| Unembedding Bias 2 | 5 |

Figure F.2: **Transformer parameters in the in-context linear regression setting**. The model has two transformer blocks for a total of $51,717$ trainable parameters.

The transformer outputs a series of tokens of the same shape as $T_k$. To read out the $\hat{y}_k$ predictions, we read out the first component of every other token, i.e.,

$$\pi_Y : \mathbb{R}^{(D+1) \times 2K} \to \mathbb{R}^K \tag{25}$$

$$\left( \begin{pmatrix} \hat{y}_1 \\ \vdots \end{pmatrix}, \begin{pmatrix} \cdot \\ \vdots \end{pmatrix}, \cdots, \begin{pmatrix} \hat{y}_k \\ \vdots \end{pmatrix}, \begin{pmatrix} \cdot \\ \vdots \end{pmatrix} \right) \mapsto (\hat{y}_1, \ldots, y_k). \tag{26}$$

### F.2.3 Training

We train from a single seed for each choice of architecture and optimizer hyperparameters using minibatch stochastic gradient descent. We train without explicit regularization and use the Adam optimizer (Kingma & Ba, 2014). The training runs take 1 to 5 TPU-hours on TPUs provided by Google Research. Models are trained from the same initialization and on the data vectors within each batch (but for different sets of tasks and task orderings).

Models are trained on a single epoch: each of the $T = 500,000$ batches consists of a new set of sequences with batch size 256. For the LLC estimates, we save 190 checkpoints: 100 are linearly spaced over the training run, and the remaining 90 are logarithmically spaced. We perform LLC estimation and other analyses on these checkpoints.

Table 4: Summary of hyperparameters and their default values for in-context linear regression transformer model training experiments.

| Hyperparameter | Category | Description/Notes | Default Values |
|---|---|---|---|
| $n$ | Data | # of training samples | 128,000,000 |
| $B$ | Data | Batch size during training | 256 |
| $T$ | Data | # of training steps | 500k |
| $N_{\text{test}}$ | Data | # of eval samples | 2048 |
| $D$ | Data | Dimensions of linear regression task (Task size) | 4 |
| $K$ | Data | Maximum in-context examples | 8 |
| $\sigma^2$ | Data | Variance of noise in data generation | 0.125 |
| $L$ | Model | # of layers in the model | 2 |
| $H$ | Model | # of attention heads per layer | 4 |
| $d_{\text{mlp}}$ | Model | Size of the hidden layer in MLP | 64 |
| $d_{\text{embed}}$ | Model | Embedding size | 64 |
| seed | Misc | Training run seeds | {0, 1, 2, 3, 4} |
| Optimizer Type | Optimizer | Type of optimizer | Adam |
| $\eta$ | Optimizer | Maximum learning rate | 0.003 |
| $\lambda_{\text{wd}}$ | Optimizer | Weight Decay | 0 |
| $\beta_{1,2}$ | Optimizer | Betas | (0.9, 0.999) |
| Scheduler Type | Scheduler | Type of learning rate scheduler | OneCycleLR |
| Strategy | Scheduler | Strategy for annealing the learning rate | Linear |
| % start | Scheduler | Percentage of the cycle when learning rate is increasing | 0.5 |

