# OpenReview forum: "Loss Landscape Degeneracy and Stagewise Development in Transformers"
_TMLR — Accepted by TMLR_

### Review · Reviewer_3d9h · 2025-04-27

**Summary Of Contributions:**

The authors study the phases of transformer learning during training using Singular Learning Theory (SLT). They posit that the learning phases observed are linked to degeneracy in the loss landscape geometry. To support this claim, they observe the degeneracy along training by computing the local learning coefficient of small transformers in language modeling and context linear regression. The authors identify distinct phases in the learning where changes in degeneracy match significant changes in the internal mechanism at play in transformers
and input/output behavior.

**Audience:**

Yes

**Claims And Evidence:**

Yes

**Requested Changes:**

I would appreciate it if the authors could discuss relevant works on phase learning and training dynamics in transformers [1-3].

[1] Edelman et al. The Evolution of Statistical Induction Heads: In-Context Learning Markov Chains. NeurIPS 2024

[2] Odonnat et al. Clustering Head: A Visual Case Study of the Training Dynamics in Transformers. arxiv 2024

[3] Chen et al. Sudden drops in the loss: syntax acquisition, Phase transitions, and simplicity bias in mlms. ICLR 2024

**Strengths And Weaknesses:**

*Strengths*

- The paper is well written
- Enough context is given on the mathematical tools, such that one does not need to be an expert on SLT to understand the main message
- I appreciate the intuitive illustrations (Figure 2/3) to convey the idea
- The experiments validate the theory well

*Weakness*

I list below a mix of remarks and questions that are not weaknesses per se, but could be clarified by the authors.

1) Several (recent) works have studied the phase learning in transformers [1-3]. Although the mathematical/methodological tools of study are different, I believe it would be relevant to discuss them in the current work.
2) Could the authors provide more details on the benefits of LLC to study the stage learning compared to the techniques used in [1-3] like visual study, loss landscape study, use of optimization tools, etc.?
3) It would be interesting to see how the method proposed by the authors scales to bigger models. I do not request such experiments, but out of curiosity, I wonder if there would be some computational issues with the SLT theory and LLC estimation. I would appreciate it if the authors could elaborate on that.

*Questions*
1) Is there any connection between SLT theory (and the LLC) and the sharpness study of the loss landscape? From Figure 2, one might think that a flat loss landscape could lead to lower LLC (e.g., first plot vs second, but several landscapes can indeed be flat but with different LLC, like the third and fourth plots).

Overall, I find the paper interesting, and the idea of applying SLT to the problem is interesting. While the benefits compared to other techniques of studies are not clear to me, I believe the paper is valuable and that the analysis could be used on other problems.

[1] Edelman et al. The Evolution of Statistical Induction Heads: In-Context Learning Markov Chains. NeurIPS 2024

[2] Odonnat et al. Clustering Head: A Visual Case Study of the Training Dynamics in Transformers. arxiv 2024

[3] Chen et al. Sudden drops in the loss: syntax acquisition, Phase transitions, and simplicity bias in mlms. ICLR 2024

---

> ### Author Response · Authors · 2025-05-21
>
> Thank you for the comments and suggestions.
>
> **Additional related works \[1-3\].** We already cite Edelman et al and Chen et al in our discussion. We were unfamiliar with  Odonnat et al and are grateful for the pointer. We have added a reference to this paper in “Related work” under “Stagewise development in deep learning.”
>
> **LLC vs other techniques**. Regarding your second question, about the benefits of LLC vs visual study, loss landscape study and use of optimization tools. The main benefit is that the LLC is derived from a fundamental theory of Bayesian learning, and as such, is significantly better motivated than e.g. the use of intrinsic dimension in Chen et al (which we note is only applied to embeddings, and therefore is not a general method for measuring model complexity suitable beyond their setting). In appendix B.5, we include a comparison against the Hessian trace in the linear regression setting. We find that the Hessian suggests additional possible substructure in stage LR2, however it misses three of the four stage boundaries discovered by the LLC. Other work by Anonymous (2025a, see top level comment) has further investigated the difference between these techniques in language models, where they also show different signals.
>
> **Scaling to larger models:** Regarding your question about larger models, please see the top level comment.
>
> **Degeneracy and sharpness/flatness:** You are right, the LLC can be thought of as precisely a measure of the flatness of the loss landscape, beyond just second order (i.e. Hessian-based measures are only sensitive to the loss to second order). The smaller the LLC, the flatter the landscape. In fact, we use *degeneracy* somewhat synonymously with what some authors might call *flatness*. In our updated LLC introduction, we have called out this connection precisely.

---

> > ### Comment · Reviewer_3d9h · 2025-05-26
> > **Thank you**
> >
> > I thank the authors for their answers.
> >
> > I first apologize for missing the Edelman and Chen citations in the first draft of your submission, and thank you for adding the other one. I do think that those works give context and might strengthen the interest of your approach. Your answer regarding the scaling to larger models is convincing, and I appreciate the clarification regarding the "flatness". For the benefits of LLC compared to other methods, as I cannot read the Anonymous 2025, I assume you discuss that in good faith, and I am convinced by your explanation. I would appreciate seeing it clearly explained in your revised manuscript, and I think it would also help the reader and practitioners decide when/why to use your approach.
> >
> > The benefits of LLC are clearer to me now, and I maintain my original comments that the current work is interesting and valuable to the TMLR audience. I thank the authors for their rebuttal and recommend acceptance.

---

### Review · Reviewer_eimZ · 2025-05-12

**Summary Of Contributions:**

This paper investigates how training dynamics in transformers reflect transitions between developmental stages. Using the Local Learning Coefficient (LLC), a metric from singular learning theory that measures local loss landscape degeneracy, the authors identify stage boundaries in two small transformer models. These stages correspond to qualitative shifts in attention patterns and task performance, such as the emergence of induction heads or robust function learning.

**Audience:**

Yes

**Claims And Evidence:**

Yes

**Requested Changes:**

- To make a more convincing case of the applicability of their theory, the authors should either include degeneracy tracking results on a larger transformer, provide a complexity analysis for computing LLC (or an approximate version) as model size scales, or make a convincing argument of why scaling up experiments is not necessary.


- It would be helpful to provide preliminary analysis or conjectures on the link between LLC drops/spikes and specific mechanistic events (e.g., neuron specialization, attention head formation).

**Strengths And Weaknesses:**

**Strengths**

- Linking loss-landscape degeneracy to model formation stages is a novel application of singular learning theory to deep transformers.
- Two contrasting settings (natural language and synthetic regression) both exhibit stagewise shifts, supporting generality across tasks. Multiple seeds, detailed appendices on LLC estimation, and comparisons to Hessian-based measures bolster confidence in the findings.

**Weaknesses**

- Estimating the LLC via SGLD is computationally intensive and demonstrated only on small (≤ 3 M-parameter) models. Only two small, attention-only architectures are studied. It is uncertain whether similar degeneracy-driven stages occur in deeper, hybrid (MLP+attention) transformers or other architectures (e.g., GPT-style models). It therefore remains unclear how this approach scales to production-scale transformers.

- While stage boundaries are empirically aligned with structural changes, the paper lacks a causal or mechanistic theory explaining why degeneracy shifts trigger specific computational structures.

---

> ### Author Response · Authors · 2025-05-21
>
> Thank you for your suggestions.
>
> **Scaling and complexity analysis.** We have added a paragraph to Appendix B.2 discussing the time- and space complexity of LLC estimation. In general, we expect the time cost to be on the order of 100s to 1000s of training steps, and the memory cost to be similar to adaptive optimizers like Adam.
>
> **On whether the *results* would scale to larger models.** In the top level comment, we addressed your concern that the LLC tracking methods would not scale to larger models. It seems you are also concerned that, assuming the methods could be scaled, our *results* may not transfer to larger models.
>
> First, let us clarify what we think of as ‘our results.’ The central claim we make with this paper is about a fundamental link between degeneracy and development. As stated in the introduction, by following the specific stage identification methodology on specific models, we find suggestive evidence of this link. The standard we accept for evaluating the scalability of our results is whether the same link is present in larger models.
>
> On these grounds, there is evidence that our results do scale to larger models. Subsequent work has provided additional suggestive evidence in larger transformers. Anonymous (2025a, see top level comment) uses a refined version of LLC tracking on Pythia-70M, a substantially larger language model, and discovers that the degeneracy reflects the formation of induction circuits in the model.
>
> Note that we do not claim that our stage identification methodology itself would be sufficient to uncover clear stages in larger models. We suspect larger models might not have such clear and isolated stages. But, by developing a more refined version of this methodology, Anonymous (2025a) appears to have been able to exploit *the same link* to study the development of a larger model through its geometry.
>
> **Link between degeneracy and mechanisms.** We agree that it is not understood exactly what mechanistic changes cause degeneracy shifts in general. However, there are several cases where we do present  connections between changes in mechanistic structure and an increase in degeneracy: see Appendix D2 on the “collapse” in layer norm layers, attention patterns, and embedding/unembedding layers of the regression transformer. . In addition, the change in LLC during the formation of the induction circuit is related to a known mechanistic factor (the circuit).

---

### Review · Reviewer_Afb3 · 2025-05-14

**Summary Of Contributions:**

This work examines the relationship between loss landscape degeneracy, as measured by local learning coefficient (LLC), and changes in transformers' internal circuits and behavior. Specifically, in two tasks (language modeling and in-context linear regression), critical points/plateau moments in the LLC score over training were identified and found to correspond to changes in model learning e.g. acquiring heuristic prediction (e.g., n-gram), ICL performance.

**Audience:**

Yes

**Claims And Evidence:**

Yes

**Requested Changes:**

- I find the discussion on the actual case studies most interesting but these sections were a bit thin compared to the background on LLC. I think it would be nice to condense the background on LLC and move some details about the actual experiments (e.g., train/eval details, the different metrics in Figs 4 and 5) up from the appendix to help interpret the results (or at least, provide more pointers to the relevant appendix sections).
- In Fig 5B, why does the model achieve better ICL for g=3 (OOD?) compared to g=1 (ID)? Does this ICL metric correspond to accuracy at all?
- It would be nice to discuss if there are more principled ways of discovering changes in model behavior/internal computation that may be reflected in the LLC changes. For the cases studied, some metrics were directly inspired by prior work, what motivated the analysis on the other metrics? Were there metrics that failed to link to any LLC changes?

**Strengths And Weaknesses:**

Strengths:
- Overall, the work presents a set of interesting results, and I appreciate that the authors examined two different task settings.
- The paper provides good context on loss landscape degeneracy and LLC for an unfamiliar audience.

Weaknesses:
- My main concern is that the relationship between LLC changes and model changes is characterized incoherently throughout the paper. The title and abstract suggest "degeneracy drives development" and that they're"deeply linked", however later in the paper it was described as "roughly coincides". I appreciate that the authors made clear that the interpretation is observational, non-exhaustive, and non-causal, but I do feel like a more in-depth discussion of the nature of this link would be needed to justify the stronger framing. For example, the authors mention being motivated by bayesian inference/singular learning process, it would be nice if these concepts can be more clearly unpacked within the context of the two particular case studies.
- It would be nice to include some analysis on checkpoints where the actual loss curve shows stage-like transition, which serves as a baseline to compare how LLC changes capture additional meaningful model changes beyond any model changes that correspond well to the actual loss.

---

> ### Author Response · Authors · 2025-05-21
>
> Thank you for the detailed suggestions and comments.
>
> **On the link between LLC and structure.** Regarding the first weakness you identify, please see the top level comment.
>
> **On stagewise development in the loss.** We include in Appendix C.1.4 a visualization of how per-token losses change across the developmental stages for the language model, and we argue that these changes are consistent with our interpretation of the stages. For example in LM3, LM4 we see stages in the LLC curve, development of induction heads in e.g. prefix score, and also the per-token losses indicate a pattern of changes consistent with induction patterns. To clarify, we do not see the value of the LLC as just being a numerical measure that reflects changes in the model beyond what can be seen in the loss curve. The underlying theory (SLT) tells us that loss and LLC are *a priori* independent quantities, and that the latter is a measure of model complexity. It is therefore a reasonable hypothesis that the LLC reflects internal structure in a way that goes beyond just tracking the loss (e.g. distinguishing memorization and generalization).
>
> **On condensing the LLC background.** As noted in the top level comment, we will take up your suggestion to condense the background on the LLC.
>
> **On Figure 5B.** Regarding your question about Fig 5B. The ICL score ICL\_{1:8} is the loss on the 8th example in-context minus the loss on the 1st example. Since the model performs better on later examples, this is negative (and more negative the better ICL is) as you note. Note that the fact that a more negative ICL score does not imply better absolute performance on later tokens. We have clarified this in an update section C.1.3. We do not understand the phenomena you are pointing to, it appears to be a nontrivial property of the algorithm learned by the transformer.
>
> **On discovering behavioral/computational changes from the loss.** Regarding the metrics that we used:
>
> * For Bigram score, Positional embedding ablation loss and N-gram score we did not provide references because these seemed sufficiently elementary to not require them. These metrics were motivated by general background knowledge about statistical structure of language, and an expectation that transformers should learn n-grams (and require positional encodings to do so).
> * The previous token score, prefix score and ICL score we adopt from earlier work (as cited).
> * In the linear regression setting, the E\[ ||y\_k^||^2\] score was motivated by our mechanistic interpretability investigations into this model (which did not make it into the final text).
> * Similarly, the layer norm analysis was motivated by these mechanistic investigations, which were too inconclusive and messy to write-up in the paper, but which gave us enough insight into the structural changes to study them with a more objective metric.
>
> The paper contains all the metrics we tried. This isn’t necessarily very indicative, because the metrics are either (a) strongly motivated by general knowledge or previous work or (b) grew out of more mechanistic investigations, so were likely by design to reflect meaningful changes in the models.

---

### Review · Reviewer_EQrd · 2025-05-16

**Summary Of Contributions:**

This paper studies how training can be stratified into distinct segments where the model is updating its internal computational structure in different ways. They propose to identify these different segments by analyzing an existing metric of the loss landscape called the Local Learning Coefficient.

**Audience:**

Yes

**Claims And Evidence:**

Yes

**Requested Changes:**

I think it would be helpful to shorten/simplify the discussion of the existing concepts like LLC and Bayesian local free energy and move the more detailed description to the Appendix. That space could then be used to present more experiments, which would also make a more compelling case for the claimed experimental findings, particularly in settings that used real world data.

Ideally, these findings and the connections they imply could be made a bit more precise by designing some sort of explicit measure of how well the curves "align". But I recognize that this could be difficult and whatever measure is chosen would necessarily be a bit arbitrary anyways. Maybe just something to think about.

Overall, I think this is a good paper and certainly worth publication in TMLR.

**Strengths And Weaknesses:**

Strengths:

* I think this paper is quite well structured. The concepts necessary to understand the method are laid out in a clear and logical order. It does feel like a lot of information to be presented just in the setup, but this seems necessary as the method relies on and builds upon several existing concepts.
* I think the Figures, particular Figure 2, are thoughtfully created and do a really good job of conveying the ideas.
* I appreciate that the paper presents experiments with both real and synthetic data.

Weaknesses:

* My primary concern with this work is the necessarily handwavy aspect of the experimental evidence. The paper's writing often seems to suggest an extremely strong connection between LLC and the network's internal structure, yet the experiments involve looking at curves and showing with shading/color that they roughly coincide. I think it would be more meaningful, for example, to try to design a metric that captures the degree to which LLC succeeds in identifying something meaningful. Say, some unsupervised learning of whether or not a model has yet achieved a particular capability, such as 2-grams, 3-grams, etc.
* I think the introduction of the concepts of LLC and Bayesian local free energy could be made perhaps a bit more high level, with a more detailed description relegated to the appendix. Alternatively, maybe some sort of indication that the details are not absolutely essential for the rest of the paper. It was sometimes a little frustrating to take the time to carefully read and understand the notation and ideas being presented, only to realize in the next section that I could have simply grasped the general idea that was being conveyed and that would have been enough for this paper's purposes.

---

> ### Author Response · Authors · 2025-05-21
>
> Thank you for your careful review and for your thoughtful suggestions. In the top level comment, we respond to your concerns about the nature of the experimental evidence, and also adopt your suggestions about moving some of the technical details to the appendices.
>
> Here, we just want to elaborate about the steps we have taken, where possible, to make the comparison between stage boundaries and specific structural/behavioral developments as objective as possible.
>
> * First, as detailed in appendix B.1, the division of training into stage boundaries is objective.
> * In the case of the bigram score, we compare the stage boundary to the minimum of the score.

---

### Author Response · Authors · 2025-05-21
**Top level comments**

We are grateful for the time each reviewer has taken to carefully read our paper, and for their thoughtful suggestions, which we feel have improved the paper. We will respond to each reviewer’s comments directly. In this top-level comment, we want to first address topics raised by multiple reviewers.

**On the strength of *claims* about links between degeneracy and development (EQrd, Afb3).** We regret that the paper seems to imply a stronger connection between the LLC and internal structure than our methodology and results support. In our revision, we have further stressed that the stage division is approximate and subject to various limitations. In particular:

* Afb3 questioned our use of the word “drives” in the title. We take your point. We have removed this.
* Afb3 also questions “deeply linked” in the abstract. We prefer to keep this line, since it is a hypothetical claim and a crucial step in the motivation of the work (see e.g. paragraph 2 in the introduction). However, based on your feedback, we realise we failed to sufficiently clarify in the introduction that this is a hypothetical claim and that we do not claim to have established such a deep link with this paper alone. We have added this clarification.

**On the strength of *evidence* about links between degeneracy and development (EQrd, eimZ).** Relatedly, there is the question of whether the evidence we contribute is a sufficient contribution.

EQrd is concerned that “the experiments involve looking at curves and showing with shading/color that they roughly coincide,” and eimZ notes that “the paper lacks a causal or mechanistic theory explaining why degeneracy shifts trigger specific computational structures.” In the respective individual responses, we point out how we have attempted to objectively judge the alignment of curves where possible, and we point to cases in the appendices where we conjecture specific mechanistic links between the changes in structure and changes in degeneracy.

Nevertheless, we think that this degree of evidence is a sufficient contribution. It is in line with the standards of evidence used in other recent works that suggest links between geometric properties of the loss landscape and changes in internal structure:

* In Achille-Rovere-Soatto “Critical Learning Periods in Deep Networks” (ICLR 2019), the main thrust of the paper is “using the information in the weights, measured by an efficient approximation of the Fisher Information, to study critical period phenomena in DNNs.” To a significant extent this reduces to showing that two curves roughly coincide (Fig 4).
* In Chen et al. “Sudden drops in the loss: syntax acquisition, Phase transitions, and simplicity bias in MLMs” (ICLR 2024, spotlight), which also uses a complexity measure (intrinsic dimension) to argue for a relation to simplicity bias over training, this is also argued for by pointing to a coincidence on a curve.

We accept that these coincidences are closer to the main argument of our paper than in Chen et al. (although perhaps not much less in Achille-Rovere-Soatto). However, we believe that our standard of evidence is at least sufficient given the TMLR criteria.

Finally, we note that subsequent work has built on the methods proposed in this paper and used this to discover new phenomena in transformer language models. These works corroborate our contribution by providing further suggestive evidence of links between degeneracy and development.

* In Anonymous (2025a), the approach of this paper was refined to study individual components of a model (attention heads) and their development, and it was shown that the LLC of an attention head correlates with degree of memorization.
* In Anonymous (2025b), ideas related to the LLC were used to discover circuits in the same small transformer model.

---

> ### Author Response · Authors · 2025-05-21
>
> **On whether the *methods* would scale to larger models (eimZ, 3d9h).** Both 3d9h and eimZ expressed concern about whether the LLC tracking method would scale to larger models than our largest (3M parameters). The answer is yes, it would scale to larger models, and in fact it has already been scaled to larger models in published work.
>
> * The paper that proposed the LLC estimator, Lau et al. (2025) “The local learning coefficient: A singularity-aware complexity measure” empirically validated the accuracy of the estimator in deep linear networks up to 100M parameters.
> * Anonymous (2025a, Appendix H), uses essentially the same LLC-tracking methodology on the open-weight model Pythia-70M (a substantially larger transformer).
>
> We have followed the suggestion of eimZ and added a discussion of the complexity cost of LLC estimation to Appendix A.2. For reference, the time and memory cost of SGLD-based LLC estimation is effectively the cost of computing a single SGD training step for the given model, multiplied by the number of SGLD steps (the number of SGLD chains times the length of the chains). For example:
>
> * In this paper, for the linear regression setting, for each LLC estimate we used 10 chains x 5000 steps \= 50k SGLD steps, comparable to 50k training steps.
> * For the larger language model, we found 20x200=4000 SGLD steps was sufficient. However, we expect this was still more than was necessary. Subsequent work (Anonymous, 2025a) used 4x200=800 SGLD steps per LLC estimate.
>
> **On the level of background detail (EQrd, Afb3, 3d9h).** Reviewer EQrd suggested streamlining the technical details of the introduction to SLT in sections 4 and 5, moving some of this material to the appendix, or advising the reader that it is not necessary to understand the definitions in full technical detail to understand the remainder of the paper. Reviewer Afb3 similarly suggested that these sections could be compressed to make more room to discuss experimental details in sections 6 and 7\. We agree with both suggestions, and have moved some details to an appendix.
>
> We hope that this change preserves the useful parts of these sections (noting that reviewers EQrd, Afb3, and 3d9h all praised the context provided in these sections). Indeed, we appreciate that SLT is not yet widely known in the community, and included this background to address that gap. However, the reviewers’ feedback has prompted us to reconsider what is essential to the main text and what can be left to appendices for readers with a deeper interest in the work.
>
> **Anonymous references.** In the above response we referred to the following anonymous papers. These papers build directly on our methodology and therefore they cite the present work. Unfortunately, this means that reading them could de-anonymise us. Therefore, in order to preserve the integrity of the double-blind review process and following the instructions of the action editor, we describe the papers and their relevant results above, but we do not provide actual citations. A minimal description of these papers is as follows.
>
> * Anonymous (2025a), a paper published in ICLR 2025 (spotlight).
> * Anonymous (2025b), a preprint posted to arXiv in April 2025\.

---

### Decision · Action_Editor_Jtrx · 2025-07-01

**Recommendation:** Accept as is

**Audience:**

Yes

**Audience Explanation:**

This paper, which applies a classically inspired measure of loss landscape degeneracy to classify high-level behavioral shifts in transformer models during training, will find an audience among those interested in better understanding learning dynamics in these models through empirical investigations. I applaud the clear, pedagogical style and the release of open-source code.

**Claims And Evidence:**

Yes

**Claims Explanation:**

The central claim of this work is a connection between the degeneracy of the loss landscape and developmental (behavioral) stages in the learning dynamics of two transformer models, as evidenced by numerical computation of the local learning coefficient (LLC; Lau et al., 2025). In response to the reviewers' requests, the revised manuscript appropriately adjusts strong claims about causality (e.g., "drives") to correlational ones (e.g., "is linked to"). With these changes, I concur with the reviewers that the claims and evidence are appropriately matched for TMLR.